# Strong influence of north Pacific Ocean variability on Indian summer heatwaves

Vittal Hari [1,2] ✉, Subimal Ghosh [3,4], Wei Zhang [5] & Rohini Kumar [2] ✉

Increased occurrence of heatwaves across different parts of the world is one of the characteristic signatures of anthropogenic warming. With a 1.3 billion population, India is one of the hot spots that experience deadly heatwaves during May-June – yet the large-scale physical mechanism and teleconnection patterns driving such events remain poorly understood. Here using observations and controlled climate model experiments, we demonstrate a significant footprint of the far-reaching Pacific Meridional Mode (PMM) on the heatwave intensity (and duration) across North Central India (NCI) – the high risk region prone to heatwaves. A strong positive phase of PMM leads to a significant increase in heatwave intensity and duration over NCI (0.8-2 °C and 3–6 days; $p < 0.05$) and vice-versa. The current generation (CMIP6) climate models that adequately capture the PMM and their responses to NCI heatwaves, project significantly higher intensities of future heatwaves (0.5-1 °C; $p < 0.05$) compared to all model ensembles. These differences in the intensities of heatwaves could significantly increase the mortality (by ≈150%) and therefore can have substantial implications on designing the mitigation and adaptation strategies.

Since the beginning of the century, there has been a significant increase in the number of mega heatwaves in India[1,2], where consequences of such events are perceived by large populations and are often met with high mortality[2–4] (Fig. 1a). In India during recent decades, heatwaves have caused more deaths than any other natural hazards[5]. India experiences deadly heatwaves during late May – June with increased mortality compared to early summer heatwaves between March and mid May (on average by four-fold increased mortality; Fig. 1a). The recent deadly heatwaves of 2015[1,6] and 2019[7], claiming more than 1200 lives occurred during late May and extended till June. June is an important month as the nation is all set to embrace the onset of monsoon which brings respite from frequent heatwaves during the drier pre-monsoon season due to rainfall induced cooling[8]. Therefore a slight delay in monsoon onset dates (5–7 days) could cause more severe and long-lasting heatwaves[9].

Despite the vulnerability to heatwaves have reduced in India over last decades[10], by virtue of increase in literacy and conversion of marginal to main working population, the overall risk has swiftly increased due to increased occurrences of such climatic hazards[4,10,11] – especially over the North Central India (NCI; Fig. 1b, indicated with a rectangular box and see Supplementary Note 2 for mapping risk to heatwaves) – one of the most prominent heatwave hot-spots in India[1,4,6,12]. The three-day maximum temperature (TXx; see Methods for more details) related to these events generally surpassed the climatological expectation by 1–1.5 °C, with more pronounced anomalies being noticed over the NCI region (Fig. 1c). Governance competencies to manage such climatic hazards are therefore impeded by several factors including, but not limited to, the large population, geographical size, accelerated haphazard urbanization[10] along with, perhaps the most important one, is a limited understanding of physical mechanisms driving such heatwave events – all these present serious challenges to the authorities for designing proper disaster preparedness strategies.

Limited studies exist aiming at deciphering the large-scale physical mechanisms responsible for heatwaves across NCI. The El Niño-

[1]Indian Institute of Technology (Indian School of Mines), Dhanbad 826004, India. [2]UFZ-Helmholtz Centre for Environmental Research, Leipzig 04318, Germany. [3]Department of Civil Engineering, Indian Institute of Technology Bombay, Mumbai 400076, India. [4]Interdisciplinary Program in Climate Studies, Indian Institute of Technology Bombay, Mumbai 400076, India. [5]Department of Plants, Soils and Climate, Utah State University, Utah, UT, USA. ✉e-mail: vittal.hari@ufz.de; rohini.kumar@ufz.de

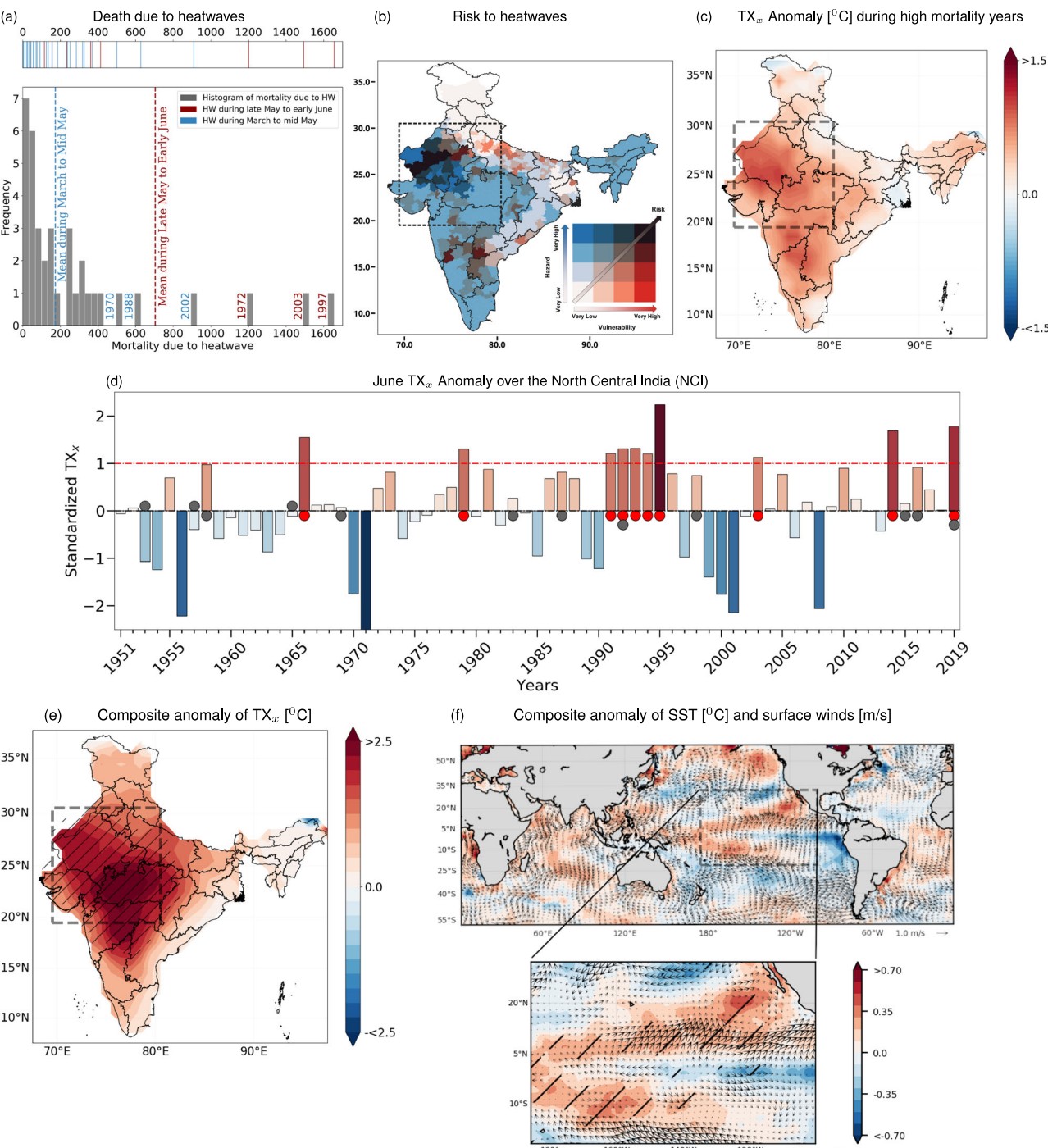

**Fig. 1 | Summer heatwave variability (June) across the North Central India (NCI) and corresponding large-scale atmospheric and sea surface dynamics.**
**a** Mortality due to heatwaves over India. Shown are the histogram of mortality to heatwaves whereas the years with major heatwaves are indicated in the top panel of **a** wherein the red (blue) colored line indicates the heatwave occurrence during late May–June (March–mid May). The red (blue) colored vertical dashed lines in the histogram represents the mean moratality coinciding during late May–June (March–mid May) periods (see Supplementary Note 1 for more details). **b** Multi-variate risk (heatwave hazard × vulnerability) map during the period 2011 over India. The 4 × 4 bivariate choropleth in the inset of map shows the classification of the risk based on the hazard and vulnerability values (see Supplementary Note 2 for the vulnerability and hazard estimations). The black rectangular box represent the high heatwave risk prone region of North central Indian (NCI). **c** Heatwave intensity (TX$_x$) anomaly for the summer (June) during deadly (high heat-related mortality)

years. **d** Inter-annual variability of the summer heatwave intensities ( °C; corresponding to the climatological mean) over the NCI region (depicted by a black rectangular region in the panel **c**). The positive (negative) anomalies are represented by red (blue) color. The red dots represent the major heatwave years wherein the TX$_x$ exceeded more than one standard deviation (indicated by the red dashed line). Black dots, on the other hand, depicts the years with strong El Niño. **e** The composite anomaly of TX$_x$ [ °C] during the major heatwave events (red dots in the panel **d**). **f** Composites of the SST ( °C) and surface wind anomalies corresponding to the major heatwave years over NCI (red dots in the panel **d**). The black rectangular box in panel **f** over the Pacific Ocean (175 °E–95 °W and 21 °S–32 °N) represents the core Pacific Meridional Mode (PMM) region. The spatial pattern of both SST and wind in this region reveal the dominant positive phase of the PMM[19] (lower panel of **f**). The hatched areas in **e** and **f** represent the locations where the anomalies are significant at the 5% level.

Southern Oscillation (ENSO) is one of the most frequently linked large-scale drivers affecting the south Asian climate variability[9,13]. Spatio-temporal variations of summer heatwaves are however not always consistent with the ENSO variability, with some of the major heatwaves occurring even during the non-prominent ENSO years[14]. Other recent studies have indicated that the heatwaves over the NCI region are due to anomalous blocking over North Atlantic Ocean[6]; and their amplification due to local land-surface and atmospheric feedback processes such as depleting soil moisture corresponding to a persistent high over the region and resulting in anomalously high temperature[1]. However, during the time-span of late May to early June – the period coinciding with occurrence of deadly heatwaves over NCI (Fig. 1a) – we neither notice a prominent blocking effect over the North Atlantic Ocean (Supplementary Fig. 1) nor the prominent influence of ENSO. Nearly 66% (two-third) of the total heatwaves occur during non-ENSO years (grey dots in Fig. 1d). While the emergence of Indian summer heatwaves and their impacts on human health (mortality) has drawn much attention in recent years[1–4,15–18], the role of large-scale atmospheric circulations along with underlying physical mechanisms associated with these events thus have still been not properly understood.

Here, using observations and climate model experiments, we detect and unravel the significant control of far-reaching tele-connection associated with the Pacific Meridional Mode (PMM)[19] on the pre-monsoon heatwaves over North-Central India (NCI) – the region with high risk to heatwaves[10] (Fig. 1b). The PMM that results from the coupled ocean and atmospheric variability over the north-eastern Pacific [175 °E–95 °W, 21 °S–32 °N; see Methods], weakens a zonal walker circulation over the Pacific, which in turn changes generally occurring lower-level westerlies into easterlies over the Indian ocean, and provides a conducive environment for hot and drier situation over NCI. Using the controlled climate model experiments (see Methods), we then corroborate the underlying mechanism through which the PMM modulates the NCI heatwave variability. Finally, we demonstrate the relevance of herein identified physical linkages for the future heatwaves projections over NCI based on the state-of-the art climate models from the Coupled Model Inter-comparison Project phase 6 (CMIP6)[20]. We find that the group of CMIP6 climate models that adequately captures the observed PMM linkages project significantly higher heatwave intensities over NCI (by almost 0.5–1 °C; $p$-value ≤ 0.05). We discuss and highlight the implication of our findings for future climate change adaptation strategies.

## Results

### Observational evidence of PMM footprint on Indian heatwaves

Here we start our analysis through observational exploration of large-scale air-sea interaction patterns including sea surface temperature (SST) and near-surface wind anomalies corresponding to heatwave variability over India. To this end, we first focus on the heatwave events which caused substantial mortality over the entire country (Fig. 1a). Thus, to further explore the possible physical mechanisms, we first identified the years with major heatwaves over the NCI region, indicated by the years with red dots in Fig. 1d; and then computed the corresponding composite anomalies of TXx (Fig. 1e) and sea surface temperature (SST; Fig. 1f). Most noticeable pattern is the systematic SST gradient along with anomalous surface circulation over the northeastern subtropical Pacific ocean, resembling the Pacific Meridional Mode[19,21] (PMM; rectangular box in Fig. 1f; see Methods). Over the PMM region, the surface wind gust towards anomalously warmer northwestern part of the tropical eastern Pacific region from the cooler region of the southeastern part. An analogous but opposite conditions of SST and surface wind anomalies are noticed during the non-heatwaves summers (Supplementary Fig. 2), which further signifies the role of PMM in modulating the inter-annual variability of TXx NCI.

To further uncover the role of PMM, we regress the TXx with the leading mode of PMM index – that characterizes the leading coupled mode of SST and surface wind over the north Pacific Ocean (see Methods) – for the observational period of 1951–2019 (Fig. 2a). In general, a positive association between PMM and TXx is noticed over a majority of the Indian region, with the strongest signal being apparent over NCI (Fig. 2a). The relationship is further assessed by taking the spatial estimates of TXx over NCI and contrasting them with the leading PMM index time-series (Fig. 2b). The inter-annual variability of the TXx and PMM index further supports the notion of significant positive association between them ($p$-value < 0.05). Significant differences in TXx over NCI are also observed between the years with strong positive (PMM+) and negative (PMM−) phases of PMM (Fig. 2c; statistically significant mean difference of 2 °C based on the non-parametric bootstrap analysis; $p$-value < 0.05). Similar observations are drawn for other heatwaves related characteristic like duration (Supplementary Fig. 3; see Methods).

### Large-scale PMM teleconnection to NCI heatwaves

In the aforementioned observational analysis, we established a significant positive association between PMM and pre-monsoon heatwaves over the NCI region. We now proceed here to understand the associated physical mechanisms for these far reaching teleconnection patterns that are often induced through changes in the large-scale atmospheric circulations[22–25]. The climatological expectations of these large-scale circulation variables (viz., outgoing long-wave radiations (OLR) – proxy for precipitation and cloud cover[6] and lower-level winds) during June exhibits a strong easterly winds at the lower tropospheric at the equatorial pacific region together with the low-level westerlies from the Indian ocean which moves towards the western pacific (Supplementary Fig. 4a). These are further supplemented with the climatology of respective (wind) velocity potential at 200 and 850 hpa levels (Supplementary Fig. 4c, e) to represent the regional convergence and divergence behaviors, both aloft and at the surface, with positive (negative) velocity potential representing a convergence (divergence) behavior. These, perhaps not surprisingly, the convergence (divergence) of winds at the troposphere along with the easterly (westerly) surface wind patterns over the eastern pacific (western pacific and Indian ocean) depicts a known walker circulation[26] – all these processes strengthen as the monsoon progresses[27].

The response of lower-level winds to the leading PMM index depicts a weakening of easterlies (westerlies) over equatorial eastern Pacific (Indian) ocean regions (Fig. 2d; see Methods for details on underlying approach). These responses are accompanied with the increased divergence (convergence) over the central-eastern tropical Pacific at the 200 (850) hpa pressure level (Fig. 2f, h). Over the western pacific region, however, the convergence at the higher level (200 hpa) is noticed, resulting in a reduced convection over the region. These (convergence/divergence) changes in easterlies/westerlies signifies the weakening of a Walker circulation pattern[22]. While we analyse the response of strength of walker circulation with PMM, a negative association is noticed (Supplementary Fig. 5), which supports the notion of significant weakening of walker circulation due to the influence of PMM. In addition, we also notice prominent (positive) OLR responses to PMM over a large swath of the NCI region and north eastern China (Fig. 2d) – indicating the region to be cloud free and with no precipitation[6] and thereby facilitating conducive conditions to enhanced heatwaves over these regions. Along with the weakening of walker circulation, the PMM forces an eastward-propagating Rossby wave similar to the stationary Rossby wave train excited by tropical SST heating anomalies[28] (Supplementary Fig. 6) influencing the changing synoptic circulations over mid-latitudes[25,28]. The similarity of the spatial pattern of wave train activities which is also to a certain degree being noticed in Supplementary Fig. 1–specifically over the mid-

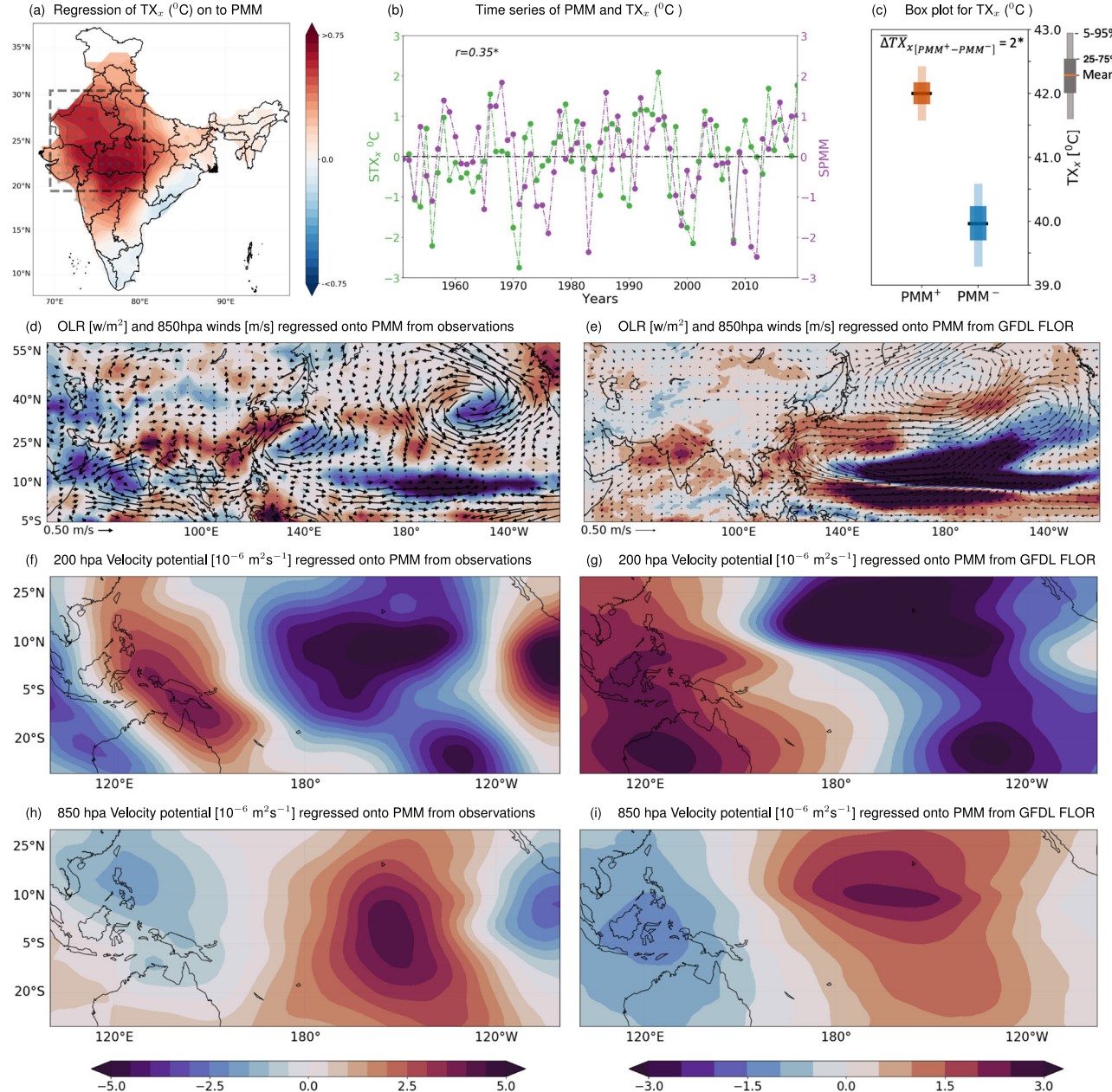

**Fig. 2 | Large-scale controls of the Pacific Meridional Mode (PMM) on the NCI summer (June) heatwave variability. a** Regression of $TX_x$ over India onto leading PMM index during the period 1951–2019. The dotted areas represent the locations where the slopes are significant at the 5% level. **b** Inter-annual variability of the leading PMM index and summer $TX_x$ estimated over the NCI region (black rectangular box in the panel **a**), with variability being represented in terms of their standardized scores. We notice a significant positive correlation between the PMM index and summer $TX_x$ variability across NCI, (*p*-value ≤0.05; represented with '*' symbol). The years with standard deviation > ( < ) 1 (−1) from this Inter-annual variability of leading PMM index is further considered as the positive (negative) phase of PMM and the $TX_x$ intensities during these phases are represented with the box plot (**c**): the differences in heatwave intensities are statistically significant (non-

parametric bootstrap analysis; *p*-value ≤ 0.05). **d** Large-scale atmospheric circulation pattern associated with the PMM in terms of the regression of outgoing longwave radiation (OLR; shading during 1975–2019) and winds at 850 hpa (vectors, during 1951–2019) from NCEP reanalysis. The response of velocity potential at 200 (**f**) and 850 (**h**) hpa levels to the leading PMM index, wherein the negative (positive) values indicates the region with increased divergence (convergence). The large-scale atmospheric circulation responses to the leading PMM index based on a long-term control GFDL-FLOR[22,38] climate model experiment (500 yr) are presented in **e, g** and **i**. A marked resemblance between observed and simulated large-scale circulation responses to the leading PMM phase over both Pacific and Indian regions, further provides the consistency of underlying PMM forcing mechanisms on NCI heatwaves.

latitudinal region. Nonetheless, their influence on NCI heatwave seems minimal (Supplementary Fig. 6).

The large-scale circulation response to PMM (Fig. 2d) partly resembles the pattern noticed during the strong positive phase of ENSO (El Niño) over the Pacific, i.e. the weakening of walker circulation[29,30] (Supplementary Fig. 7). The role of ENSO in modulating the Indian heatwave variability is highly debatable[9,13,31] and there is an

ambiguity in attributing such events to ENSO[1,6,10,14,29]. This notion is also affirmed in our analysis – a weaker correspondence between the El Niño years and $TX_x$ variability over NCI (Fig. 1d). In contrast, our analyses show a significant association between the PMM and the NCI heatwave characteristics (intensity/duration) even accounting for the confounding role of ENSO (Supplementary Figs. 7a and 8). To further confirm this notion of PMM on the NCI heatwaves, we performed a

climate perturbation experiment. For this purpose we use atmospheric general circulation model developed by International Centre for Theoretical Physics (ICTP AGCM)[32,33], which could properly capture the observed pattern of climatology over India[34] (See Supplementary Note 3 for more details and Supplementary Fig. 9a, c).

The positive phase of PMM forcing resulted in a increased outgoing longwave radiation (OLR) over the NCI region and weakening of westerlies at 850 hpa (Supplementary Fig. 9b); and this bears a strong agreement with the observations (Fig. 2). While we saw the difference in the response between the PMM and El Niño to the large-scale circulations, we notice more pronounced increased in the OLR – specifically over NCI and also a substantial weakening of westerliers (Supplementary Fig. 9d), which indicates that the effect of PMM in more pronounced than El Niño in causing the heatwaves over NCI region. Additionally, studies reported that PMM is coupled with ENSO through a wind-evaporation-SST feedback mechanism[35], sometime even acting as a precursor and trigger of ENSO[36]. PMM can also occur independent of ENSO or it can jointly act to persuade the variability in the North Pacific[37]. Nonetheless, our analysis for the neutral ENSO years further asserted a statistically significant ($p$-value < 0.05) linkage between PMM and NCI heatwaves characteristics (Supplementary Fig. 8).

To further ascertain the modulation of PMM on the NCI heatwaves, we complement the above observational analysis with controlled climate model experiment. To this end, we use the 2500 year-long simulations of the fully (atmosphere-land-ocean) coupled Geophysical Fluid Dynamics Laboratory Forecast-Oriented Low Ocean Resolution (GFDL-FLOR) performed under controlled conditions of radiative forcing and land parameters corresponding to a pre-industrial level[22,38] (see Methods for more details). The GFDL-FLOR based simulations are able to capture the observed climatological features of large-scale circulation patterns representing the walker circulation pattern (Supplementary Fig. S4). The pronounced PMM control on the weakening of easterlies (westerlies) over eastern pacific (Indian) ocean regions along with the increased divergence (convergence) over the eastern pacific at different pressure levels is also evident in the control GFDL-FLOR model simulations (Fig. 2e, g, i and Supplementary Fig. S10). Moreover, the substantial increase in an OLR is noted over the Indian region, indicating PMM teleconnections imposes significant heatwave conditions over NCI. Albeit the greater resemblance of these model-based responses to those of observational ones (Fig. 2d, f, h), we note that the climate simulations have a slightly different response (magnitude-wise) of large-scale circulations, which among other things could be due to imprints of anthropogenic warming conditions in contemporary observational datasets[39–41], differences in spatial resolutions and different forcing conditions or model errors or/and biases

## The responses of moisture transport and monsoon onset to the PMM

The onset of the Indian summer monsoon that generally happens during June has significant influence on the water resources and agricultural sectors of India[42,43]. It also marks the termination of pre-monsoon heatwaves due to evaporative cooling[9]; and any significant delay in the onset of monsoon can result in exceptional and deadly heatwaves over India[8]. Majority of moisture source during the onset period generally originates from the Indian Ocean – of which the Arabian Sea is the most dominant contributing region[44,45] (Fig. 3a). The PMM responses, on the other hand, weaken the Walker circulation and impairs the westerlies that result in reduced eastward moisture flow into the Indian mainland from the contributing major moisture sources (Fig. 3b). These processes have a substantial impact on the progression of monsoon rains from the north Indian Ocean towards the India mainlands, for example through a delay in Monsoon onset dates and thereby increased intensities of heatwaves across NCI (Fig. 3c).

We notice a significant delay in the monsoon onset dates between years with the prominent positive and negative phases of PMM (Fig. 3c; on-average by ≈7 days with $p$-value of <0.05; see Methods for details on estimation of the onset dates). Additionally, the total number of rainy days during June is also reduced over NCI (Fig. 3d) as well as several other regions across India including western ghats, north India and core monsoon zone (Supplementary Fig. 11). Contrasting responses are however noticed in the leeward side of the western ghats, due to anomalous moisture in-surge from Bay of Bengal (Supplementary Fig. 11). Nevertheless, over the NCI region – where PMM has a significant influence on the heatwave characteristics – a profound difference in the rainy days are observed between the years with strong positive and negative phases of the PMM (Fig. 3d; difference of ≈1.5 days estimated based on the non-parametric bootstrap analysis with $p$-value < 0.05). This significant delay in the monsoon onset suggests that the extreme heat waves could, in part, be a product of local heating, along with its connection to the large-scale atmospheric circulation. The delay in monsoon onset drops the soil moisture to anomalously low levels, increasing the net surface radiation which leads to significant positive sensible heat flux. This could further leads to a substantial land surface feedback that exert additional control on the development of heatwaves across Indian region[17]. Nonetheless, the changes in climate responses to PMM – delay in the monsoon onset dates and reduced rainy days during the prominent positive PMM phase, and vice-versa – provides additional evidences on the role of PMM teleconnections in modulating the heatwaves across the NCI region.

## Implications of adequate representation of PMM in the CMIP6 models/ensembles

In the aforementioned analysis, we have established the potential linkage between the PMM and NCI pre-monsoon heatwave characteristics. Here, we analyse the state-of-the-art climate models from the CMIP6[20] simulations in representing such linkages and their implications on the future projections heatwave characteristics over NCI. To this end, first we contrast the observed spatial pattern of leading PMM expansion (SST) coefficients over the core PMM region[19,22,46] with the corresponding ones obtained from each of 31 analyzed CMIP6 model simulations during the historical period 1951–2014 (See Methods; see also Supplementary Table 1 for CMIP6 models). In general, the CMIP6 climate models are able capture the spatial pattern of this large-scale coupled mode of SST and surface wind reasonably well (Fig. 4a, b) with a mean rank correlation ($\rho$) of 0.55 (see Methods; and Supplementary Table 1 for individual model skill) which is in line with the previous assessments[47]. Along with the individual CMIP6 models, we also tested the capability of capturing the PMM with the use of large ensemble of a historical simulations and a role of internal variability (See Supplementary Note 4 and Supplementary Fig. 12 for more details) and shows that almost two thirds of the ensembles showing the response of PMM to NCI temperature similar to that of the observations – indicating that the internal variability does not eclipse the variation in the response of PMM towards the NCI temperature variability.

We then identify the group of CMIP6 models/realizations that not only adequately represent the spatial structure of leading (SST) expansion coefficients but also the observed linkages between the PMM and heatwave characteristics over NCI (number of models/realization = 6 with correlation ($\rho$) ≥ 0.50 and the PMM response of higher TXx intensity during its positive phase over NCI; see Methods and Supplementary Table 1 for more details). The leading SST coefficient pattern over the core PMM region based on these models/realizations (Set A) not only resemble the observations quite well, but also exhibit considerably reduced inter-model spread compared to that of all model ensembles (Fig. 4c). We notice almost two to three fold reduction in the inter-model spread – one standard deviation values depicted as contours in Fig. 4b, c – which are statistically significant ($p$-

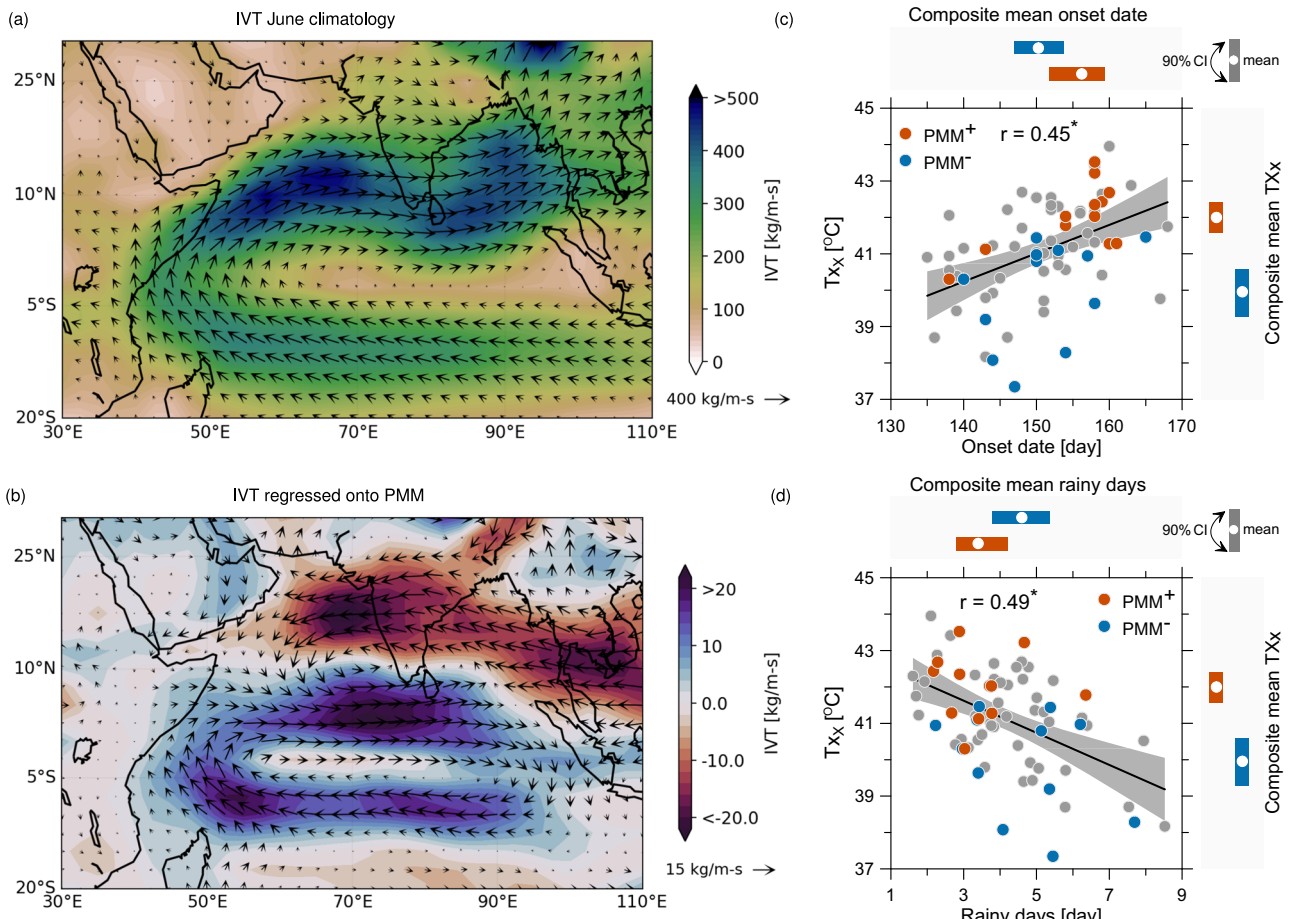

**Fig. 3 | Implications of the large-scale patterns associated with the Pacific Meridional Mode (PMM) towards the moisture transport and onset of Indian monsoons. a** Characteristics of June climatology of moisture transport, in terms of integrated vapor transport (IVT)[72] during the period 1951–2019. Majority of moisture source for India during June (monsoon onset period) is generally from the north Indian Ocean–of which the Arabian Sea region is the dominant source[44,45]. **b** The response of IVT to leading PMM index shows the inhibition of moisture transport to the Indian mainland from the adjacent north Indian ocean due to weakening of westerlies. The consequences of this is perceived in the relationship between $TX_x$ and monsoon onset days (**c**) based on Hydrologic Onset and With-drawal Index (HOWI) (see Methods and Supplementary Fig. 16 for methodological details), which is statistically significant at 5% level. The black line represent the linear fitting between $TX_x$ and onset days, while the gray shaded region depicts the confidence interval of the linear fitting. The bars on the top (right) represent the differences in onset dates ($TX_x$) between the prominent PMM phases. **d** is same as **c** but for relationship between $TX_x$ and rainy days.

value ≤ 0.05; estimated based on a non-parametric bootstrap sampling technique taking into account the differences in sample sizes between all models and Set A model/realization; see Methods). In addition, the Set A models/realizations also represents a better response of weakening of Walker circulation to PMM compared to other group models/realizations (Supplementary Fig. 13; shown in terms of the OLR, winds and velocity potential at lower pressure level).

Next, we contrast the projected changes in pre-monsoon heat-wave intensity (TXx) over NCI based on the two groups of CMIP6 models (all model ensemble vs Set A models/realizations) across different Shared Socioeconomic Pathways (SSPs[48]; see Methods). In general, the CMIP6 models project hotter conditions during the second half of the 21st Century ($\overline{\Delta TX_X}$; 2065–2100) over NCI compared to the contemporary historical estimates (1980–2014) – on average by 2–4 °C higher TXx under moderate to higher emission scenarios (SSP2-4.5–SSP5-8.5; Fig. 4d). Here the riveting aspect is significant differences in future TXx projections between the two groups of CMIP6 models (Fig. 4d) – the Set A models/realizations project on average $\overline{\Delta TX_X}$ 0.5–1 °C (p-value ≤ 0.05; see Methods) higher compared to all model estimates under the moderate to high emission scenarios (SSP2-4.5–SSP5-8.5). This notion is even evident in the large-scale circulation patterns, with the Set A models/realizations able to capture the observed patterns, i.e., weakening of walker circulation

(Supplementary Fig. 13). Similar differences are noticed for projections of average temperature ($T_{avg}$) for considering all individual CMIP6 models (Fig. 4e) and the large-ensemble (Supplementary Fig. 14; See Supplementary Note 4 for more details).

These differences for the $\overline{\Delta TX_X}$ and $\overline{\Delta TX_{avg}}$ over NCI between two model/realization groups are substantial considering that the heat-wave intensities observed during the contemporary high mortality heatwave years are on average around 0.7 °C higher than normal conditions (Fig. 1a, c). To further contextualize this aspect, a recent study[2] developed a novel probabilistic model of heatwave related mortality using temperature as climate predictor, reported that even moderate increase in mean temperature (≈0.5 °C) would lead to a significant rise in heatwave related mortality (by ≈150%) across India. Such intense heatwaves that can create significant disruptions in functioning of socio-environmental systems such as agricultural productivity, ecological activities, human health, among others[2,3,8,9,11] is within a range of the projected temperature differences ($\overline{\Delta TX_X}$ or $\overline{\Delta TX_{avg}}$ 0.5–1 °C) noticed here between the two groups of CMIP6 models (Fig. 4d, e). As such these signify the importance of herein identified PMM linkages in adequately constraining climate model projections of heatwaves across NCI.

We emphasis that the main finding of our study is the far-reaching impact of PMM and its novel association to the extreme temperatures

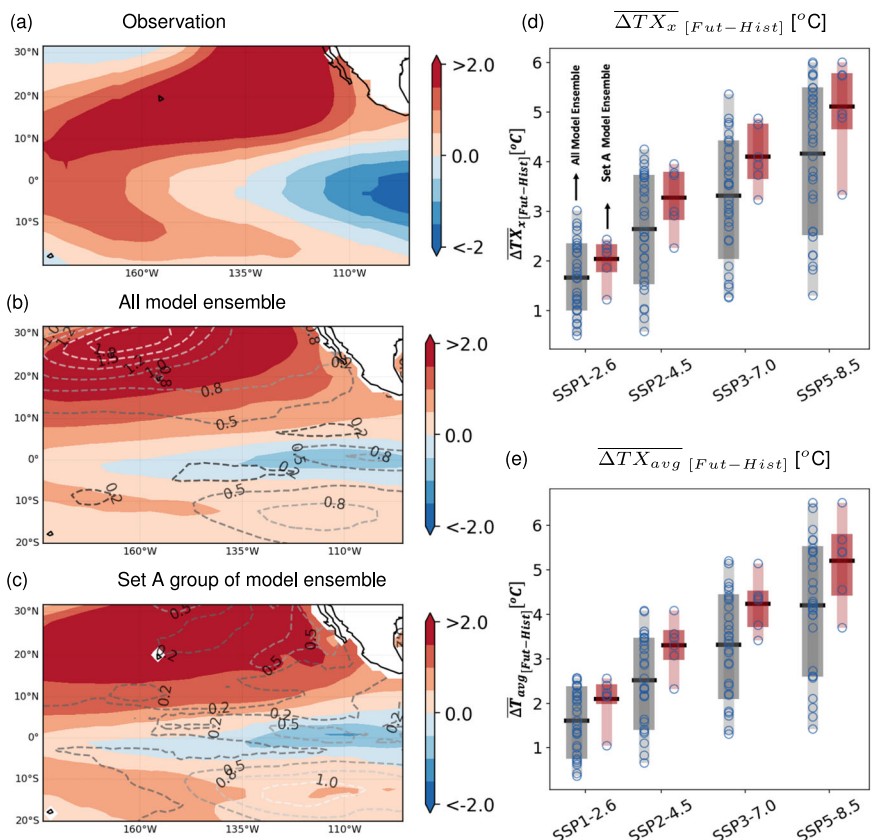

**Fig. 4 | Representation of Pacific Meridional Mode (PMM) in the state-of-the-art CMIP6 climate model simulations and their implications on the future projections of heatwaves.** The leading coupled mode of SST and surface winds over the core PMM region, obtained from the maximum co-variance analysis (see Methods). For depiction, we show the spatial patterns of SST expansion coefficient for **a** observational data and the ensemble mean of all **b** and Set A model/realization simulations (**c**). The contours in **b** and **c** represents the standard deviation of the SST leading mode amongst the ensemble members. Refer related texts in Methods for more details on the criteria for grouping the considered CMIP6 models. The

individual model skills are provided in Supplementary Table 1. **d** Box-plots depicting the distribution of projected changes in the TXx intensities under different SSP over NCI region during the second half of the twenty-first century (2065–2100) w.r.t. the historical simulations (1980–2014). The blue open circles show individual model realizations considered for each groups. The Set A model/realization project a systematically higher heatwave intensities over NCI (by approximately 0.5–1 °C). Similar observations have been made even when we analyzed projected changes in the average surface temperature over NCI region (**e**).

over the northern Indian region. Along with their overlying large-scale atmospheric circulation pattern, we demonstrate these associations through both observations and climate model based control and perturbation experiment analyses. As a part of implication aspects, we then proceed with showing the usefulness this novel association for the extreme temperature projections from CMIP6 climate model projections. To this end the idea was to see if we can see the coherent and consistent differences in future temperature projections between different group/ensemble of climate model simulations. We indeed find and report that a set of models (ensembles) that better capture the observed PMM response, have significantly different (higher) projected changes in temperature extremes over India as compared to the rest of the models/ensembles. As to why such differences lie between two set of models/ensembles – we hint this to their abilities to capture observed PMM response – and certainly this would require a more detail and comprehensive study to further investigate looking for example underlying individual model physically mechanisms and/or tracking a trajectory of particular realization.

## Discussion

The observational historical hot summers, fueled by anthropogenic warming[49–53], have caused substantial socioeconomic and environmental damages across India[2,3,8,9,11]. The projections from the state-of-the-art CMIP6 climate model simulations generally indicate a increased intensities of heatwaves during summer (Fig. 4d); and such

heatwaves are going to further intensify, unless the emissions are properly controlled[10,31] (Fig. 4d). Despite all climate models project an increasing heatwave intensities, the overall magnitude of the projected increase is subject to a large uncertainty. For developing effective mitigation and adaptation strategies, the decision-makers would require a reliable projections from the climate models that should adequately represent observed signatures of dominant large-scale climate modes[54]. In this context, our study establishes a signifying role of PMM in modulating the heatwave characteristics over India (NCI) – the robustness and consistency are established with using both the observational analysis and the control climate model experiment results. Further, these substantial linkages reveal that a substantial bias in climate models leads to an significant ($p$-value $\leq 0.05$) underestimation of heatwave intensities in the future. Overall, our study underscores the necessity of proper representation of these large-scale dynamics in current generation of climate models, whose value to the adaptation and mitigation strategies cannot be stressed enough[2,9,55–57].

## Methods

### Datasets and observational analysis

The assessment of heatwave characteristics (intensity and duration) over NCI from 1951–2019 is performed using gridded daily maximum temperature data from India meteorological department[58], which is available at a 1° spatial resolution. The rainfall data is also procured

from the same institute[59], however, it is available at a spatial resolution of 0.25°. The heatwave intensities (TXx) are estimated using a three-day $T_{max}$ annual June maximum temperature determined from the daily $T_{max}$ data[60]. In addition, we estimate the heatwave duration ($TX_{days}$) – defined when the daily $T_{max}$ exceeds the 90th percentile value of all daily $T_{max}$ values in the June. We chose the period 1961–1990 to estimate the percentile value based on the suggestion of Expert Team for Climate Change Detection and Indices (ETCCDI)[61–63]. The atmospheric patterns underlying the association between PMM and NCI heatwaves are procured from the National Centers for Environmental Prediction (NCEP)/National Center for Atmospheric Research (NCAR) reanalysis project (NCEP-NCAR-1)[64] for the period 1951–2019 available at 2.5° resolution. The SST data used in the present study are obtained from sea ice and sea surface temperature (HadISST) dataset produced by the Met Office Hadley Centre[65] at 1° spatial resolution. Although we focus on the June month (pre-monsoon period), the overall results remained same even when we analyzed the responses of PMM to heatwave characteristics considering both May and June months (Supplementary Fig. 15).

The PMM is a leading mode of variability in the northeastern subtropical Pacific characterizing the coupled mode of SST and surface winds[19]. It is defined as the first maximum covariance analysis (MCA) mode of coupled SST and surface winds in the Pacific Ocean. Before the application of MCA, the seasonal cycle is removed, data are detrended, and later a three month running mean is applied to the data, and the equatorial Pacific Cold Tongue Index is removed. PMM generally describes the meridional variations in SST, winds and convection in the tropical Pacific[19,21,22] (Fig. 4). The PMM has been proposed to primarily originating from the North Pacific Oscillation (NPO), which is characterized by a north-south oscillation in sea level pressure over the North Pacific[66]. It should be noted that the ENSO effect is removed through linear regression onto the cold tongue index (CTI) when calculating the PMM index[19,22]. Along with the initiation and development of ENSO, the PMM can also alter the mid-latitude and equatorial atmospheric circulation significantly[19], and thus lead to changes in the magnitude of precipitation[23,67,68], temperature[69] and even tropical cyclones[22,23,70] over these regions. However, its influence on the variability of summer heatwave characteristics and their associated mechanisms are still unidentified. Following the previous studies[19,22], the PMM index and the leading SST expansion co-efficient are computed based on the observational SSTs and surface winds from the National Centers for Environmental Prediction-National Center for Atmospheric Research (NCEP-NCAR) Reanalysis[64]. Monthly estimates of the PMM index can be obtained from the NOAA Earth System Research Library, available at https://www.esrl.noaa.gov/psd/data/timeseries/monthly/PMM/ for the period 1948–present.

## Control climate model experiment

Along with the observational data, here we use a fully-coupled long-term Geophysical Fluid Dynamics Laboratory (GFDL) Forecast-Oriented Low Ocean Resolution (FLOR) coupled climate model control experiments, with the radiative forcing and landuse of the 1860 level, to understand the PMM response to the NCI heatwaves variability. This model has been widely used to examine the climate system with a spatial resolution of ~50km for the land and atmosphere components[22,38]. The control experiments are performed for a total of 2500 years, however, here we focus on the 500 years for the analysis of PMM and NCI heatwave for the sake of computing efficiency. The main purpose to use this experiment is to test whether such a PMM-NCI heatwave association also holds for the long-term control simulation of GFDL-FLOR, which would provide a dynamical framework in which to understand the physical causes of this connection[22]. The PMM index in the control experiment is also calculated based on the method proposed in Chiang and Vimont[19].

## Monsoon onset estimation

To estimate the onset, we use the Hydrological Onset and Withdrawal (HOWI)[71]. HOWI is derived by considering the dynamics of moisture flux through the computation of vertically integrated moisture transport (VIMT) over the Indian Ocean, including Arabian Sea (5°N–20°N and 45°E–80°E), with the use of NCEP-NCAR-1 reanalysis data. The VIMT averaged over the selected region is normalized by the following transformation to get HOWI:

$$\chi' = 2[\chi - \min(X')]/[\max(X') - \min(X')] - 1 \qquad (1)$$

where X′ is the mean annual cycle and χ′ is the normalized VIMT time series. Noticeable variation in the HOWI, especially during the onset and the withdrawal phase of Indian summer monsoon, is observed from the mean annual cycle (Supplementary Fig. 16a). The onset day is further defined as the day when the HOWI turns positive, as we demonstrate in Supplementary Fig. S16a. The selected approach agrees well with the onset dates provided by the IMD (Supplementary Fig. 16b).

## Future heatwave projections based on CMIP6 simulations

To quantify the impacts/importance of PMM in the future projection of heatwaves, the mechanism established between the summer PMM index and heatwave characteristics is further evaluated using the state-of-the-art global climate model simulations from the Coupled Model Intercomparison Project phase 6 (CMIP6)[20] (model descriptions are provided in Supplementary Table 1). The climatological spatial patterns of some basic parameters such as precipitation and sea level pressure for all the selected models matches well with the observations for these variables (Supplementary Fig. 17), indicating the capability of these models in representing the basic tropic states. In case of future projections, we utilize four shared socioeconomic pathways (SSPs), drawn from Tier 1 of ScenarioMIP[48]: SSP1-2.6 (+2.6 W m⁻²; low forcing sustainability pathway), SSP2-4.5 (+4.5 W m⁻²; medium forcing middle-of-the-road pathway), SSP3-7.0 (+7.0 W m⁻²; medium- to high-end forcing pathway), and SSP5-8.5 (+8.5 W m⁻²; high-end forcing pathway). These SSPs are classified based on the assumptions of economic and population growth, investment in health and education, and climate mitigation efforts[48]; and also on the different possible range of future greenhouse gas emissions.

Here, we first evaluate the individual CMIP6 model ability to capture the observed spatial pattern of the leading SST expansion coefficient (positive PMM phase) over the Pacific. Similar to observational analysis, we first remove the seasonal cycle, detrend the data and apply a three month running mean for both variables. Next, an equatorial Pacific Cold Tongue Index (SST averaged over 180°W–90°W, 6°S–6°N) is removed from all the grid points on both SST and wind fields to limit the ENSO influences[19]. Finally we perform the MCA analysis based on the covariance matrix formed between SST and surface winds. The covariance matrix is decomposed through Singular Value Decomposition (SVD), and the leading statistical mode form the PMM SST patterns. We consider the 65 years (1951–2014) historical model simulations for contrasting the spatial characterization of the leading PMM expansion coefficients between observations and CMIP6 model simulations. In case of future scenario, we follow the similar method for obtaining the PMM index, but for the period 2065–2100 (Supplementary Fig. 18).

Next we analyze the NCI heatwave intensities response to the PMM from the individual CMIP6 model simulations (see Fig. 2). For this purpose, we estimate the differences in averaged TXx over NCI between the years with prominent positive phase of PMM and rest of the years (Supplementary Table 1 for more details) and identify those model simulations, which could represent the prominent warmer

condition during the positive PMM phase. Based on these information, we classify the group of CMIP6 model as Set A model/realization which could simultaneously fulfill the criteria of adequately: (1) representing the spatial features of leading expansion coefficients ($\rho \geq 0.50$), and (2) capturing the PMM response to the NCI heatwave $\overline{\Delta TX_X}$ differences between the composites of during $PMM^+ - PMM^-$ is $\geq 0.5\,°C$). The PMM states in the future scenarios showed that they could able capture the spatial pattern of the coupled mode of SST and surface wind reasonably well (Supplementary Fig. 18). Further, for each CMIP6 climate model, we compute the relative changes in the future TXx across different SSPs for the period (2065–2100) compared to the respective historical simulations (1980–2014). We then contrast these projected changes between the two model groups – i.e., all vs. Set A model/ realization (Fig. 4d and refer Supplementary Fig. 19 for their spatial characteristics). We also test the sensitivity of the $\rho$ threshold values ($\rho \geq 0.55$ and $0.60$) and find that our main finding still holds good on the reported increased $\overline{\Delta TX_X}$ in Set A model/realization of CMIP6 models (see Supplementary Fig. 20). In addition, when we look into the number of positive phases between the Set A model/realization and rest of models, we notice that the positive phases are substantially greater in the Set A model/realization compared to the rest of the CMIP6 models (Supplementary Fig. 18) – which may lead to increased temperature in the Set A model/realization.

### Statistical significance

The statistical significance of the difference in mean of heatwave characteristics between the years with strong positive and negative PMM phases are estimated using a bootstrap analysis. The bootstrap analysis are gaining a lot of interests among researchers to estimate the significance level of certain statistical analysis as it does not make any assumptions pertaining to distribution and sample size of the data. It is thus suitable for a number of circumstances including for comparing distributions of different sizes, as is the case here. We re-sample (with replacement) each of the PMM phases heatwave characteristics (intensity/duration) for 10,000 realizations. We then computed the number of times, out of these 10,000 samples, the warmer condition is apparent during the positive PMM phase compared to the composite of either the negative PMM phase or the climatological mean; and this corresponding ratio represents the non-parametric $p$-value estimate.

Similar analyses are performed to estimate the significance level on the increase in projected $\overline{\Delta TX_X}$. For this purpose, we re-sample all CMIP6 model ensembles for 10,000 times with a sample size similar to the Set A model/realization (varying between five and seven depending on the threshold criterion to classify adequate models). For each realization, we then estimate the respective statistics – specifically mean – and compared against that of the Set A Models. We then compute the corresponding p-value based on the number of times in the re-sampled data, the mean statistic fall to the the corresponding mean of the group of Set A model/realization.

### Data availability

The daily maximum temperature data are available from https://www.imdpune.gov.in/Clim_Pred_LRF_New/Grided_Data_Download.html. The NCEP–NCAR data are available from https://psl.noaa.gov/data/gridded/data.ncep.reanalysis.html, the CMIP6 data from https://esgf-node.llnl.gov/projects/cmip6/. The PMM index is obtained from the NOAA Earth System Research Library, available at https://www.esrl.noaa.gov/psd/data/timeseries/monthly/PMM/. The GFDL FLOR control runs can be procured from https://nomads.gfdl.noaa.gov/dods-data/FLOR/CM2.5/CM2.5_A_Control-1860_FLOR_B01_FA_HAD13_iv/pp/atmos/ts/monthly/20yr/. Other processed datasets can be made available upon reasonable request from the corresponding authors.

### Code availability

The codes can be procured from VH, upon a request.

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

## Acknowledgements

This work was partly carried out within the bilateral project XEROS (eXtreme EuRopean drOughtS: multimodel synthesis of past, present and future events), funded by the Deutsche Forschungsgemeinschaft (grant RA 3235/1-1; RK and VH) and Czech Science Foundation (grant 19-24089J; RK and VH). The authors also acknowledge efforts of different organizations/people for making the data available for this work which include CMIP6 data from the World Climate Research Programme's Working Group on Coupled Modelling; historical climate reconstruction, IMD, NCEP and GFDL-FLOR control simulation datasets.

## Author contributions

V.H. and R.K. conceptualized and designed the study with inputs from S.G. W.Z. helped with the climate model experimental simulations. VH conducted the analysis with inputs from S.G., W.Z., and R.K. V.H. and R.K. drafted the manuscript.

## Funding

## Competing interests

The authors declare no competing interests.
