## [Peer Review File · Nature Communications]

Strong influence of north Pacific Ocean variability on Indian summer heatwavesREVIEWER COMMENTS

Reviewer #1 (Remarks to the Author):

Authors construct an relationship between heatwaves in North Central India (NCI) and Pacific Meridional Mode (PMM) over the North Pacific on interannual timescale. Most part of the manuscript describes possible mechanism that how PPM could impact the NCI region. In the final part, they try to apply the interannual relationship on constraining future projection of NCI heatwave. However, there exist some big flaws that make the either mechanism or constraint less valid. Substantial revision is necessary if not rejected. Specific comments are seen below.

Major comments:

1. L65-68: Authors use Supp. Fig. S1 to show no Atlantic blocking effect on the NCI heat wave, as well as ENSO. However, In Fig. S1, there indeed exist strong wave activities between 30N-60N, including the North Atlantic, and they seem being related to the anomalous anticyclone over the NCI. What's more, although authors do not show the anomalous SST during the NCI heatwave, the anomalous tropical winds exhibit a pattern of weakened Walker circulation that indicates an El Nino phase. So, I do not think what author claim here is valid.
2. L68-69: I noticed that most gray dots correspond to the years when TXx index does not exceed 1sigma, and even very negative, except for two years (1992 and 2019). I do not think they are heatwaves.
3. L151-152 : I do not think it is simply a pattern of weakened Walker circulation. As shown in Fig. 2f and 2h, the anomalous VP shows like a wave train. It is indeed ascending in the central Pacific and descending in the western Pacific. However, in the eastern Indian Ocean is upward motion as well. So, it can not be treated as a simple or conventional weakened Walker circulation.
4. L152-155:"an anomalous cyclonic pattern and negative OLR to PMM responses over the core PMM region" Where is the core PMM region? There indeed is an anomalous cyclonic pattern between 25N and 45N, but it is far away from the so-called PMM region shown in Fig. 1f, the amplified box, in which the anomalous SST centers around 20N. It is more important that the anomalies far away from the equator cannot be drawn an analogy with classical Matsuno-Gill response.
5. L160-162: Please show the pattern during El Nino for reader to compare with the patterns here.
6. L83-187: More significant differences between modeled (Fig. 2e,h,i) and observational anomalies should be highlighted, such as strong negative heating near the equator, no ascending motion over the eastern Indian Ocean. These differences could lead to different mechanisms.
7. L90-191: "which among other things could be due to imprints of anthropogenic warming conditions in contemporary observational data-sets". Why not use large ensemble of a historical simulation?
8. The most questionable is the last part that tries to constrain projection using the relationship between PMM and NCI heatwave. Usually, such a relationship on interannual timescale is not stable, which is strongly modulated by internal variability. We cannot select so-called "adequate models" based on random noise. Whether the relationship between PPM and NCI in observations and all models are stable should be fully examined. Second, most important factors determining the projection uncertainty of heatwave in NCI may not the relationship with the PMM on interannual timescale but others such as climate sensitivity. Because timescale of projection is multi-decadal to century, the controlling factors are often different from that on interannual scale. All of these reasons make the constrained results unreliable.

Minor comments:

1. L100 and in all related figures: I suggest using TXx (without a subscript and a different font for the last 'x'), keeping the format consistent with IPCC AR5 (Chapter 2 Box 2.4, 2013).

2. Statistical significance of all anomalous patterns in Figures should be tested.
3. Figure 2a: I cannot find "The hatched areas".
4. L133-144: I cannot understand why suddenly it is turned to use a chunk of words to describe the climatology over the Indo-Pacific region, from the previous sentence said "we first examine the ... patterns ...associated with the PMM variability". It should be deleted.
5. L229: Delete the first "we".

Reviewer #2 (Remarks to the Author):

**Review comments for Manuscript#: NCOMMS-21-12439-T
(North Pacific Ocean variability constrains the future projections of heatwave in India)**

This manuscript shows a relationship between June higher heatwave risk over North Central India (NCI) and Pacific Meridional Mode (PMM). Positive PMM mode weakens the tropical Pacific Walker Circulation, leading to an anomalous stable condition over northern India and decrease in moisture transport into India. In the last part, the authors argued that in the CMIP6 models, models that have better performance in simulating the historical connection between +PMM and NCI heatwaves also project a warmer June three-day temperature maximum (TX_{-x}) in the future, implying the North Pacific Ocean variability contributes to the future projection of heatwave in India.

Although I've found it is convincing +PMM can impose warmer and drier condition and delay the monsoon onset in June over northern India according to the historical records, there are at least two major issues in the manuscript that are critical. As a result, I do not think the manuscript has reached the standard for publishing on Nature Communications. The comments are elaborated below:

1. The title and the last part of the manuscript strongly suggest that the future projections of increased intensity of heatwaves in northern India is due to PMM, but the results are not enough to support the argument. First, the authors selected the "adequately represented models" based on historical records. How would the PMM or the North Pacific change in the different emission scenarios? Would the projected change in the Walker Circulation due to the PMM be consistent with the projected change in Indian temperature in June?

Also, could it be because those "adequately represented models" have better representation in the tropical basic-state? The projected change in the TX_{-x} might be due to the change in the tropical basic-state. Models that have better representation in the tropical basic-state could also be able to better capture the historical relationship between the +PMM and NCI TX_{-x}.

2. The definition of heatwave is confusing and not enough justified. In the Abstract and the beginning of the Introduction, the authors emphasized that heatwaves that happened in late May to early June had higher mortality (Fig. 1a). Then, they used June TX_{-x} to define the heatwave events. It seems like the high mortality of "late May to early June" heatwaves was dominated by three events (1972, 2003, 1997). Further, 1972 and 1997 were not listed in Fig. 1d based on "June" TX_{-x}. Is it because one was focusing on "late May to early June" and one was focusing on "June"? Alternatively, is it possible that "duration" of heatwaves is also critical to the mortality? Besides this inconsistency, there are two follow-up questions regarding "duration":

(1) I am not sure why the authors did not consider "duration" when defining heatwave events since they did not justify their selection of the heatwave definition. They cited Meehl et al. 2004 (Science) when applying TX_{-x} as their criteria to define the heatwaves. However, in Meehl et al. 2004, they also considered another criterion that including "duration" threshold. Their results suggested that heatwaves in North America were projected to be longer, supporting that "duration" is also an important element to be considered.

(2) There is a section (L193-L223) that shows +PMM could delay the onset of monsoon in June and suggests that the delay of the monsoon onsets can result in exceptional heatwaves over India (L197-198). I don't think the results are supportive to this argument, especially given the definition of heatwaves in this study is based on June TX-x. The results or description did not explicitly connect the delayed onset of monsoon to higher TX-x intensity. I would think that a delayed onset of monsoon could extend an existing heatwave (duration matters), or increase the number of high three-day temperature (TX), or increase the probability of experiencing higher TX-x (due to land-atmosphere coupling?).

3. The results from GFDL-FLOR need to be checked. Fig. S4 c-f, climatology from GFDL-FLOR are very off, compared to observations. It seems like the latitude is flipped. Also, in the regression plots (Fig. 2g compared to Fig. 2f), 200hPa velocity potential over Indo-WP is quite different from the observations, would this make any substantial difference in the surface responses over India?

4. The authors suggested that El Nino does not have relationship (or very weak) with NCI June TX-x. El Nino also has reduced OLR anomalies over the tropical central Pacific (similar to Fig. 2d), weakening the Walker Circulation. Could the authors elaborate more why El Nino has weak or no relationship with India?

5. Statistic significant test should be applied to the composites shown in Fig. 1ef and regression shown in Fig.2d-i.

6. Fig. 2d-i: the latitude and longitude ranges are different in these panels, which makes it quite difficult to compare these figures. I would suggest to use the same longitude range for every panel, and indicate the equator.

7. Fig. 4 b,c: I am not sure what those contour represent.

Reviewer #1:

Authors construct an relationship between heatwaves in North Central India (NCI) and Pacific Meridional Mode (PMM) over the North Pacific on interannual timescale. Most part of the manuscript describes possible mechanism that how PPM could impact the NCI region. In the final part, they try to apply the interannual relationship on constraining future projection of NCI heatwave. However, there exist some big flaws that make the either mechanism or constraint less valid. Substantial revision is necessary if not rejected. Specific comments are seen below.

Response #1: We thank the Reviewer for appreciating our work. We have followed all the suggestions. Addressing those has certainly improved the quality of manuscript significantly.

Major comments:

1. L65-68: Authors use Supp. Fig. S1 to show no Atlantic blocking effect on the NCI heat wave, as well as ENSO. However, In Fig. S1, there indeed exist strong wave activities between 30N-60N, including the North Atlantic, and they seem being related to the anomalous anticyclone over the NCI. What's more, although authors do not show the anomalous SST during the NCI heatwave, the anomalous tropical winds exhibit a pattern of weakened Walker circulation that indicates an El Nino phase. So, I do not think what author claim here is valid.

Response #2: We thank the Reviewer for this important comment. We agree with the Reviewer at this point that there is some indeed exist a wave activities between 30N-60N. With regard to PMM, it is noted that it forces an eastward-propagating Rossby wave similar to the stationary Rossby wave train excited by tropical SST heating anomalies [1]. Fig. R1a displays the regression map of 250-hPa wind fields and geopotential height over the entire Northern Hemisphere (NH) with the PMM and this shows a strong wave activities – specifically in the mid-latitudes. To ascertain more on this, we further estimate the wave activity flux (WAF) to understand the energy propagation pathways. It is considered as a diagnostic tool for intensifying the propagation, source, and sink of a propagating packet of a stationary or migratory quasi-geostrophic wave[2]. It is usually estimated at the 250 hpa pressure level and is given by the following equation[2];

$$\mathbf{W} = \frac{p \cos \varphi}{2|\mathbf{U}|} \left(\begin{array}{l} \frac{\mathbf{U}}{a^2 \cos^2 \varphi} \left[\left(\frac{\partial \psi'}{\partial \lambda} \right)^2 - \psi' \left(\frac{\partial^2 \psi'}{\partial \lambda^2} \right) \right] + \frac{V}{a^2 \cos \varphi} \left[\frac{\partial \psi'}{\partial \lambda} \frac{\partial \psi'}{\partial \varphi} - \psi' \frac{\partial^2 \psi'}{\partial \lambda \partial \varphi} \right] \\ \frac{U}{a^2 \cos \varphi} \left[\frac{\partial \psi'}{\partial \lambda} \frac{\partial \psi'}{\partial \varphi} - \psi' \frac{\partial^2 \psi'}{\partial \lambda \partial \varphi} \right] + \frac{V}{a^2} \left[\left(\frac{\partial \psi'}{\partial \varphi} \right)^2 - \psi' \left(\frac{\partial^2 \psi'}{\partial \varphi^2} \right) \right] \\ \frac{f_0^2}{N^2} \left\{ \frac{U}{a \cos \varphi} \left[\frac{\partial \psi'}{\partial \lambda} \frac{\partial \psi'}{\partial z} - \psi' \frac{\partial^2 \psi'}{\partial \lambda \partial z} \right] + \frac{V}{a} \left[\frac{\partial \psi'}{\partial \varphi} \frac{\partial \psi'}{\partial z} - \psi' \frac{\partial^2 \psi'}{\partial \varphi \partial z} \right] \right\} \end{array} \right) \quad (1)$$

where \mathbf{W} denotes the WAF ($\text{m}^2 \text{s}^{-2}$), p is pressure, ψ is the stream function, $\mathbf{U} = (U, V)$ is horizontal wind vector, with U and V denoting the zonal and meridional wind components, respectively; λ and φ are longitude and latitude, respectively; f_0 is the Coriolis parameter; N^2 is the Brunt-Vaisala frequency; a is Earth's radius and the prime represents the anomaly.

Fig. R1b shows a PMM response of streamfunction (shading) and WAF

(vectors) which further confirms a distinct wavelike structure over the subtropics of NH. This wave structure is consistent with the high and low centers of the circumglobal teleconnection (CGT) pattern, which corresponds to the second leading empirical orthogonal function of the inter-annual variability of the upper-tropospheric circulation over NH midlatitudes. Luo et al.[3] reported a significant correlation between PMM and CGT index, which is defined as the areal mean of the 200-hPa geopotential height averaged over 35° – 40° N, 60° – 70° E[4]. These results suggest that the SST gradient pattern in eastern subtropical North Pacific (i.e., warming in the northwest and cooling in the southeast of eastern subtropical North Pacific) can trigger a propagation of Rossby wave over the NH midlatitudes. These observations are strikingly similar to what we noticed in Fig. S1 and these details are now provided in the revised manuscript (now provided as Supplementary Fig. S6 in the revised manuscript).

Figure R1: Large-scale atmospheric circulation pattern associated PMM in terms of the of geopotential height (shading) and wind (vectors) at 250-hPa level (a). Spatial patterns of the wave activity flux (WAF; vectors) and streamfunction (shading) at 250-hPa level, depicting the prominent feature of circumglobal Rossby wave propagation during the leading PMM mode (b).

We agree with the Reviewer that the large-scale circulation response to PMM partly resembles the pattern noticed during the strong positive phase of ENSO over the Pacific, i.e. the weakening of walker circulation. Nonetheless, Our analyses show a significant association between the PMM and the NCI heatwave characteristics (intensity/duration) even accounting for the confounding role of ENSO (Supplementary Fig. S8; these aspects were already discussed in the manuscript). To further concertize the role of PMM on the NCI heatwaves, now we performed a climate perturbation experiment. For this purpose we use atmospheric general circulation model developed by International Centre for Theoretical Physics (ICTP AGCM)[5, 6], which could properly capture the

observed pattern of climatology over India[7].

To evaluate the impacts of the SST anomaly on NCI heatwaves and large-scale circulation, we performed two sets of perturbation experiments which are integrated for 30 years. Initially, we start the experiment by prescribing the seasonal climatology of SST based on HadISST (CLIM). Further, to evaluate the response of NCI heatwaves, the observed composite anomaly of SST during positive phase of PMM (Fig. R2a) and El Niño (Fig. R2c) for June are superimposed on the seasonal climatology, keeping SST identical to the CLIM experiment for other months. The subtraction of CLIM experiment with the SST anomaly forced experiment then provides the response of large-scale atmospheric circulation, to both PMM and El Niño.

Figure R2: Sea surface temperature (SST) anomalies used in ICTP AGCM perturbation experiment for both PMM (a) and El Niño (c) phase. (b) Response of outgoing longwave radiation (OLR; in shading) and 850 hpa wind (vectors) to the perturbation experiment forced with PMM sst anomalies. These changes are estimated based on the seasonal climatology runs (CLIM). The differences in the response between the PMM and El Niño is provided in (d).

The positive phase of PMM forcing resulted in a increased outgoing longwave radiation (OLR) over the NCI region and weakening of westerlies at 850 hpa (Fig. R2b); and this bears a strong agreement with the observations (Fig. 2 and 3 of the main manuscript). While we saw the difference in the response between the PMM and El Niño to the large-scale circulations, we notice more pronounced increased in the OLR – specifically over NCI and also a substantial weakening of westerlies (Fig. R2d), which indicates that the effect of PMM is more pronounced than El Niño in

causing the heatwaves over NCI region. These aspects are now provided in the revised manuscript (now provided as Supplementary Fig. S9 in the revised manuscript). Additionally, Studies reported that PMM is coupled with ENSO through a wind–evaporation–SST feedback mechanism[8], sometime even acting as a precursor and trigger of ENSO[9]. PMM can also occur independent of ENSO or it can jointly act to persuade the variability in the North Pacific[10]. However, these aspects needs to be evaluated in details and can be considered as a possible future research.

2. L68-69: I noticed that most gray dots correspond to the years when TXX index does not exceed 1sigma, and even very negative, except for two years (1992 and 2019). I do not think they are heatwaves.

Response #3: The gray dots depicts the years with strong El Niño, not the years corresponding to the NCI heatwave years. This was already mentioned in the caption. We notice only two heatwave events occurred during the strong ENSO years (i.e., nearly 80% of heatwaves over NCI occurred during non-ENSO years). These results are in line with the results obtained by ICTP experiments, where we show a limited influence of ENSO on the June NCI heatwave characteristics.

3. L151-152: I do not think it is simply a pattern of weakened Walker circulation. As shown in Fig. 2f and 2h, the anomalous VP shows like a wave train. It is indeed ascending in the central Pacific and descending in the western Pacific. However, in the eastern Indian Ocean is upward motion as well. So, it can not be treated as a simple or conventional weakened Walker circulation.

Response #4: We thank the Reviewer for this comment. PMM is known to have a teleconnection pattern to several regional climatological characteristics through changes in both lower and upper-tropospheric circulation changes[11]. In the lower level, a prominent anomalous cyclone is generated as a Gill-Matsuno-type response to increased diabatic heating associated with the warming component of PMM[11] and their effects have been documented in changing the characteristics of summer precipitation extremes over China[11] and tropical cyclones in the Western North Pacific[12]. The upper-tropospheric circulation changes, on the other hand, are characterized by an eastward Rossby wave train (Fig. R1), with their effects have been noticed on causing the summer heatwaves and precipitation extremes over China[3, 11]. Therefore, we agree with the Reviewer with this point that other than weakening of Walker circulation, PMM does influence the triggering of eastward Rossby wave train (Fig. R1).

Moreover, we do believe that PMM response of weakened walker circulation may be the preliminary cause for the changes in the characteristics of the NCI heatwaves. The eastward propagation of Rossby wave train also have some influence on the increased geopotential height and streamfunction over Indian region, however, their influence is limited (as seen in the Fig. R1). To emphasize more the influence of PMM on the weakening of walker circulation, we performed an additional analysis (Fig. R3). To this end, we analyse the relationship between strength of

walker circulation (SWC) with the PMM during June. Here, the SWC is estimated by sea level pressure anomaly differences across the Pacific, which is associated with vertical motions of the walker circulation[13]. The described region over the tropical Pacific is 4.74°S – 4.74°N in latitude, and 128.39°E – 151.05°E and 211.47°E – 231.61°E in longitude for the western and eastern tropical Pacific edge, respectively. In general, a negative association between PMM and SWC is noticed (Fig. R3a), which supports the notion of significant weakening of walker circulation due to the influence of PMM. Further, a significant differences in SWC are also observed between the years with strong positive (PMM⁺) and negative (PMM⁻) phases of PMM (Fig. R3b; statistically significant mean difference of -0.40 mbar based on the non-parametric bootstrap analysis; p -value < 0.10). These results are further confirms the role of PMM in weakening of walker circulation and this information is now provided in the revised manuscript (now provided as Supplementary Fig. S5 in the revised manuscript).

Figure R3: Relationship between strength of walker circulation with the PMM during June. (a) Inter-annual variability of the leading PMM index and strength of walker circulation (SWC), with variability being represented in terms of their standardized scores. We notice a significant negative correlation between the PMM index and SWC, (p -value ≤ 0.10 ; represented with “*” symbol). The years with standard deviation $>$ ($<$) 1 (-1) from this Inter-annual variability of leading PMM index is further considered as the positive (negative) phase of PMM and the SWC during these phases are represented with the box plot (b): the differences in SWC are statistically significant (non-parametric bootstrap analysis; p -value ≤ 0.10).

4. L152-155: “an anomalous cyclonic pattern and negative OLR to PMM responses over the core PMM region” Where is the core PMM region? There indeed is an anomalous cyclonic pattern between 25°N and 45°N , but it is far away from the so-called PMM region shown in Fig. 1f, the amplified box, in which the anomalous SST centers around 20°N . It is more important that the anomalies far away from the equator cannot be drawn an analogy with classical Matsuno–Gill response.

Response #5: We agree with the Reviewer that a prominent low-level cyclone is seen over the western North Pacific responds to the increased diabatic heating over the northeastern subtropical Pacific caused by the

warming SST anomalies. This response of warming SST anomalies could be identified as a Gill-Matsuno-type[3, 11, 12]. In addition to this, recently another mechanism named "summer deep convection" (SDC) was proposed by Amaya et al.[14] in connection to these anomalous warming of SST over the Pacific. By restoring SST anomalies in a coupled model to the observations in the North Pacific, the authors found that the mean Inter-Tropical Convergence Zone (ITCZ) is sensitive to the PMM in the boreal summer and fall as it moves northward along its seasonal cycle. As a result, PMM-related SST anomalies can generate enhanced deep convection during these seasons, which produces Gill-like atmospheric circulation anomalies throughout the subtropical North Pacific[12]. Such an atmospheric response directly results in zonal surface wind anomalies over the equator, which provides another pathway for the subtropical SST anomalies to influence the deep tropics. All these studies, along with ours, points that the PMM can force a Gill-like response that projects onto the equator.

5. L160-162: *Please show the pattern during El Niño for reader to compare with the patterns here.*

Response #6: We have now provided these plots in the revised manuscript (now provided as Supplementary Fig. S7 in the revised manuscript).

6. L83-187: *More significant differences between modeled (Fig. 2e,h,i) and observational anomalies should be highlighted, such as strong negative heating near the equator, no ascending motion over the eastern Indian Ocean. These differences could lead to different mechanisms.*

Response #7: We thank the Reviewer and agree that there is some differences between the observations and GFDL FLOR control experiments. These are now highlighted in the revised manuscript. As mentioned earlier, there are many mechanism involved with the PMM, of which the changes in the Walker circulations[12] and eastward propagation of Rossby wave[3] is well documented; and also been noticed in the present study. All these information is now provided in the revised manuscript.

7. L90-191: *"which among other things could be due to imprints of anthropogenic warming conditions in contemporary observational data-sets". Why not use large ensemble of a historical simulation?*

Response #8: We thank the Reviewer for this comments. To address this, we have now utilized the large-ensemble available from the MPI grand ensemble (MPI-GE) and MIROC6 CMIP6 climate model. The basis of the selection of this particular model is mainly based on the fact that MPI-GE offer the best global and regional representation of both the internal variability and forced response in observed historical temperatures compared to other CMIP5 large ensembles[15]. On the other hand, we select MIROC6 CMIP6 large ensemble because it generally falls under the observational constrained group of models. The MIROC6 has a climate sensitivity values that are within the IPCC AR5 likely range and also shows that warming trend is much more consistent with the observations, compared to other available large ensemble such as CanESM5[16].

Here first we analyse the PMM and their associated with the NCI heat-waves using piControl experiment. For this purpose, we utilize the 500 years of control simulation for both MPI-GE and MIROC6. We notice that the piControl experiment from both the large-ensemble models could capture the prominent coupled meridional mode over the pacific (Fig. R4a and d). While we analysed the summer PMM index and their association with the OLR and surface winds over the south Asia and we find a stark similarities as we noticed from the observations (Fig. 2). There is an apparent increase in the OLR and weakening of the westerlies, which may be conducive for the increased temperature over India, specifically over NCI (Fig. R4b and e). Further, while we analyse the response of temperature over India to summer PMM index, we indeed noticed an increased temperature over north India, including NCI region (Fig. R4c and f).

Figure R4: **Representation of PMM in the piControl runs of large ensembles from MPI grand ensemble (MPI-GE) and MIROC6 CMIP6 climate model** (a) The leading coupled mode of SST and surface winds over the core PMM region, obtained from the maximum co-variance analysis for 500 years of piControl simulations for MPI-GE. (b) Relations of OLR and surface winds; (c) surface temperature with the summer PMM index for the 500 years piControl simulation of MPI-GE. (d, e, f) is similar to (a, b, c) but for the 500 year piControl simulation of MIROC6 CMIP6 climate model.

Further to address the aspect of internal variability, we perform the similar analysis as we did with the piControl experiment. Here we utilize all the available 100 (50) ensemble members of MPI-GE (MIROC6) and see the individual ensemble's capability in reproducing the PMM pattern. Fig. R5a and d shows that most of the ensemble do present the prominent mode of PMM with minimal standard deviations across the ensemble members (shown in contours). Further, the response of summer PMM to the OLR and surface winds also been preserved in the most of the ensembles, with almost two-third of the ensembles showing increased

OLR over the Indian region (Fig. R5b and e; shown in strips). Because of this, we further notice a substantial increase in temperature, specifically over NCI (Fig. R5c and f). Thus, internal variability does not eclipse the variation in the response of PMM towards the NCI temperature variability from both the large-ensemble models. All these discussions are now provided in the revised manuscript (now provided as Supplementary Fig. S12 in the revised manuscript).

Figure R5: **Representation of PMM in the historical runs of large ensembles from MPI grand ensemble (MPI-GE) and MIROC6 CMIP6 climate model** (a) Shows the ensemble mean 100 members of the MPI-GE historical simulations for the leading coupled mode of SST and surface winds over the core PMM region during historical time period, i.e., 1951–2014. The contours represents the standard deviation of the SST leading mode amongst the ensembles members. (b) and (c) depicts the ensemble mean of the response of OLR, surface winds and surface temperature with the summer PMM index, respectively for MPI-GE. The stippling denotes at-least two-third (66%) of the 100 ensemble members agreement in signs. (d,e and f) is similar to (a, b and c) but for 50 member of MIROC6 large ensemble.

8. *The most questionable is the last part that tries to constrain projection using the relationship between PMM and NCI heatwave. Usually, such a relationship on interannual timescale is not stable, which is strongly modulated by internal variability. We cannot select so-called “adequate models” based on random noise. Whether the relationship between PPM and NCI in observations and all models are stable should be fully examined. Second, most important factors determining the projection uncertainty of heatwave in NCI may not the relationship with the PMM on interannual timescale but others such as climate sensitivity. Because timescale of projection is multi-decadal to century, the controlling factors are often different from that on interannual scale. All of these reasons make the constrained results unreliable.*

Response #9: We thank the Reviewer for this comments. To address this particular comment of the Reviewer, we follow the method employed by the Huang et al.[17] in evaluating the historical PMM index and NCI temperature. It should be noted that for this purpose we use only MIROC6

since MPI-GE showed that all the ensemble did represent the relationship between summer PMM and NCI temperature equally good – with very less internal variability in the future projections as well. Therefore, for MIROC6 50 ensembles, we identify that how could the PMM constraints the future projections of temperature change over NCI region of India. As already shown in Fig. R5f that most of the ensemble members could reproduce the increased temperature response to summer PMM index. This is even true for the area averaged temperature over NCI, with most of the ensembles are having a positive response to PMM (Fig. R6a). From these ensembles, we select 10 members with the highest trends of temperature with PMM – which is near to the observed trend, and analyse the effect of PMM on the future projections (represented by the red circles in Fig. R6a). We notice a significant increase in the surface temperature (approximately 0.15 – 0.17 degree C) in these 10 ensembles compared to other ensembles (Fig. R6b). These results are re-confirming the assertion that the models which capture both the PMM patterns and its association with the temperature variability over NCI during the historical period, projects higher temperature compared to those models which don't. All these discussions are now provided in the revised manuscript (now provided as Supplementary Fig. S14 in the revised manuscript).

Figure R6: **Evaluation of the historical PMM and NCI temperature in large ensemble** (a) The NCI temperature (boxed region as shown in Fig. R5c and f) relation with the summer PMM index for the observed (black circle) and the 50 MIROC6 members (blue circles) during 1951-2014. The 10 members with the higher positive association with the PMM index are shown in red circles. (b) Box-plots depicting the projected changes in the surface temperature intensities under SSP 5-8.5 scenario over NCI region during the second half of the twenty-first century (2065–2100) w.r.t the historical simulations (1980–2014) for the 10 members with the higher positive association with the PMM (red colored) and for the rest of the members (blue colored).

Minor comments:

1. L100 and in all related figures: I suggest using TXx (without a subscript and a different font for the last 'x'), keeping the format consistent with IPCC AR5 (Chapter 2 Box 2.4, 2013).

Response #10: Thanks for the suggestion. We have now accordingly changed this notation in the revised manuscript.

Statistical significance of all anomalous patterns in Figures should be tested.

Response #11: We thank the Reviewer. We have now provided the statistical significant test results, where-ever it is necessary, in the revised manuscript.

3. Figure 2a: I cannot find “The hatched areas”.

Response #12: We have now rectified this in the revised manuscript.

4. L133-144: I cannot understand why suddenly it is turned to use a chunk of words to describe the climatology over the Indo-Pacific region, from the previous sentence said “we first examine the ... patterns ... associated with the PMM variability”. It should be deleted.

Response #13: Done

5. L229: Delete the first “we”.

Response #14: Done

Reviewer #2:

This manuscript shows a relationship between June higher heatwave risk over North Central India (NCI) and Pacific Meridional Mode (PMM). Positive PMM mode weakens the tropical Pacific Walker Circulation, leading to an anomalous stable condition over northern India and decrease in moisture transport into India. In the last part, the authors argued that in the CMIP6 models, models that have better performance in simulating the historical connection between +PMM and NCI heatwaves also project a warmer June three-day temperature maximum (TXx) in the future, implying the North Pacific Ocean variability contributes to the future projection of heatwave in India.

Although I've found it is convincing +PMM can impose warmer and drier condition and delay the monsoon onset in June over northern India according to the historical records, there are at least two major issues in the manuscript that are critical. As a result, I do not think the manuscript has reached the standard for publishing on Nature Communications. The comments are elaborated below:

Response #1: We thank the Reviewer for appreciating the quality of the work and our efforts towards recognizing the large-scale connectivity in explaining the inter-annual variability of heatwaves over NCI. We have made a sincere effort to address all the concerns raised by the Reviewer in the revised manuscript, and addressing those certainly have improved the quality and clarity of the presented work – without changing the main conclusion of the article which we have proposed in our earlier submission.

1. The title and the last part of the manuscript strongly suggest that the future projections of increased intensity of heatwaves in northern India is due to PMM, but the results are not enough to support the argument. First, the authors selected the “adequately represented models” based on historical records. How would the PMM or the North Pacific change in the different emission scenarios? Would the projected change in the Walker Circulation due to the PMM be consistent with the projected change in Indian temperature in June? Also, could it be because those “adequately represented models” have better representation in the tropical basic-state? The projected change in the TXx might be due to the change in the tropical basic-state. Models that have better representation in the tropical basic-state could also be able to better capture the historical relationship between the +PMM and NCI TXx.

Response #2: We thank the Reviewer for this important comment. We agree in line with the Reviewer that our manuscript attempts to understand the un-identified role of the PMM in causing the heatwaves over India – specifically over northern Indian region – along with their associated large-scale atmospheric circulation pattern. We believe the results from the observations and climate model control/perturbation experiment as a important and main results of our manuscript. The CMIP6 simulations were merely used – as a secondary information – to shed some light on the implication aspects of the results obtained from the observations. Although, we have now – after addressing the comments from Reviewers – substantially improved the CMIP6 simulation results, we believe more analysis is required to pinpoint the discrepancies in the

CMIP6 simulations and their representation of the PMM and its response to subcontinent climatic condition, which can be considered as a future research scope. Therefore, considering all these factors, we have now changed the title of our manuscript to *”Strong influence of north Pacific Ocean variability on Indian summer heatwaves”*.

Figure R7: Representation of PMM in the state-of-the-art CMIP6 climate model simulations for different future scenarios. The leading coupled mode of SST and surface winds over the core PMM region, obtained from the maximum co-variance analysis. For depiction, we show the spatial patterns of SST expansion coefficient for (a) ensemble mean of adequate models and (b) ensemble mean of rest of the models for the medium emission scenario, SSP2-4.5. (c) and (d) is same as (a) and (b), but for the high emission scenario, SSP5-8.5. (e) The comparison of the PMM index for the future period, i.e., 2065–2100 for two emission scenarios (SSP2-4.5 and 5-8.5) are shown in the form of probability distribution function (PDF). (f) Depicts the box plot representing the difference in number of positive phases between adequate and rest of the models. The differences are significant at 5% level.

To begin with, We analyse the response of PMM state in different emission scenarios during the period 2065–2100. For this purpose, we considered two representative pathways with medium (SSP2-4.5) and higher emission (SSP5-8.5) scenarios. Fig. R7a and b shows that, in general, the SSP2-4.5 scenario could able capture the spatial pattern of this large-scale coupled mode of SST and surface wind reasonably well. The noticeable

aspect here is that the group of adequate models (Fig. R7a) has the intensity of SST patterns higher than the adequate set of models (Fig. R7b). Further, we performed a similar analysis for SSP5-8.5 (Fig. R7c and d) and noticed an homogeneous results as with the SSP2-4.5 – with the adequate models having increased intensity (Fig. R7c) compared to the rest of the considered models (Fig. R7d). Overall, our analysis suggests that in the future scenarios, the intensities of PMM index is higher in the SSP5-8.5 compared to the SSP2-4.5 (as shown in PDF in Fig. R7e), with the adequate models are showing higher intensities of SST pattern over PMM region compared to the rest of the models. Further, when we look into the number of positive phases between the adequate and rest of models, we notice that the positive phases are substantially greater in the adequate models compared to the rest of the CMIP6 models (Fig. R7f) – which may lead to increased temperature in the adequate models, as shown in Fig.4 of the main manuscript.

Figure R8: Basic states of the CMIP6 models in terms of velocity potential at 850 hpa pressure level and their responses to PMM both in historical and future simulations. (a) Ensemble mean of climatology of velocity potential at 850 hpa level for the ensemble mean of adequate group of models during the historical period (1951–2014). (c) and (e) shows the ensemble responses of velocity potential to PMM index during the historical and future period, respectively, for adequate group of models. The future period considered here is from 2065–2100 and for the SSP5-8.5 scenario. (b), (d) and (f) is same as (a), (c) and (e) but for rest of the CMIP6 models.

Further – on the basic state of the climate models – we performed an analysis by considering velocity potential (VP) at 850 hpa level. Fig. R8a

shows that the climatology of VP for the historical period for the group of adequate models, which depicts the positive (convergence) over western pacific and negative (divergence) over the eastern pacific – which is similar to that of observed pattern of climatology (Supplementary Fig. S4) representing a typical walker circulation[18]. For the rest of the models also, we notice a similar pattern (Fig. R8b) which indicates that there is no difference in representing the basic state (or climatology) between the two sets/group of models. Similar observations have been made when we considered sea level pressure and precipitation over the tropics (Fig. R9). The climatological spatial patterns matches well with the observations for these variables, indicating the capability of these models in representing the basic tropic states.

The responses of VP to the PMM during the historical period (Fig. R8c) further shows an increased convergence over the eastern pacific at 850 hpa pressure level. Over the western pacific region, the divergence at the lower level is noticed, resulting in a reduced convection over the region. These (convergence/divergence) changes signifies the weakening of a Walker circulation pattern[12] – which is strikingly similar to the observed responses of the PMM to the VP (as shown in Fig. 2 of the main manuscript). During the future scenario, we notice a further weakening of walker circulation with additional increase in convergence/divergence over the eastern/western pacific at 850 hpa pressure level – indicating further increased weakening of walker circulation (Fig. R8e). The rest of the models – on the other hand – shows an increased convergence (divergence) in western (eastern) pacific, which is quite opposite to what we have noticed both in the observations and in the group of adequate models (Fig. R8d) and this disparities further increases in the future simulations as well (Fig. R8f). Overall, this analysis shows that the selected group of adequate models could represent the observed PMM responses to VP and this response will substantially increase in the future – which is not depicted in the group of rest of models. This could be the reason for the adequate models in having an increased temperature extremes in the future compared to the rest of the models. All these discussion are now provided in the revised manuscript (now provided as Supplementary Figs. S13, S17 and S18 in the revised manuscript).

2. The definition of heatwave is confusing and not enough justified. In the Abstract and the beginning of the Introduction, the authors emphasized that heatwaves that happened in late May to early June had higher mortality (Fig. 1a). Then, they used June TXX to define the heatwave events. It seems like the high mortality of “late May to early June” heatwaves was dominated by three events (1972, 2003, 1997). Further, 1972 and 1997 were not listed in Fig. 1d based on “June” TXX. Is it because one was focusing on “late May to early June” and one was focusing on “June”? Alternatively, is it possible that “duration” of heatwaves is also critical to the mortality? Besides this inconsistency, there are two follow-up questions regarding “duration”:

Response #3: We thank the Reviewer for this comment. With regard to definition of heatwaves, we would like to mention that there is no consistent and methodological approach[19, 20]. Several criteria have been utilized to characterize the heatwaves based on either mean, maximum,

Figure R9: Basic states of the CMIP6 models in terms of surface pressure and topical rainfall during the month of June. (a) and (b) is the climatology of surface pressure and precipitation for the NCEP NCAR reanalysis data. (c) and (d) is the climatology for the set of adequate models. Shown are the ensemble mean of the climatology from the set of models. (e) and (f) are for the set of rest of models.

minimum temperature, humidity, or a combination of all these[21]. In this study, we selected the TX_x criteria, estimated using a three-day T_{max} annual June maximum temperature determined from the daily T_{max} data[22] – which in a way inherently considers the duration in their characterization. Nonetheless, considering the concern of the Reviewer, we have now considered another well known heatwave indices, i.e., TX90P criterion. More details on this are further provided in the following comment of the Reviewer.

Furthermore, with Fig. 1a, we wanted to emphasize that the most devastating heatwaves occurred over India during late May and early June. This conclusion is irrespective of the regions within India. As for example, the 1997 the late May – early June heatwave event occurred over the east coast of India[23]. We have performed this analysis to show the importance of late May – early June heatwave events in India. The Fig. 1d, on the other hand, shows the TX_x anomalies for the NCI region, where the risk (i.e., vulnerability and hazard) to heatwave is usually higher compared to any other regions of the nation (Fig. 1b). In addition, although we focus on the June month (pre-monsoon period), the overall results remained same even when we analysed the responses of PMM to heatwave characteristics considering both May and June months (Supplementary Fig. S15).

(1) I am not sure why the authors did not consider “duration” when defining heatwave events since they did not justify their selection of the heatwave definition. They cited Meehl et al. 2004 (Science) when applying TX_x as their criteria to define the heatwaves. However, in Meehl et al. 2004, they also considered another criterion that including “duration” threshold. Their results suggested that heatwaves in North America were projected to be longer, supporting that “duration” is also an important

element to be considered.

Response #4: We have already considered the “duration” in our analysis. Supplementary Figure S3 shows the regression of June heatwave indicator in days (estimated with the 90 percentile threshold, estimated by considering the base period of 1961–1990) over India onto June PMM index during the period 1951–2019. Similar observations, as with TXx, are also drawn for this heatwaves related characteristics, with NCI region being positively associated with the PMM.

Figure R10: Regression of June heatwave indicator, i.e., TX90P (warm days; percentage of days when daily maximum temperature exceeds 90th percentile), over India onto June PMM index during the period 1951–2019. The hatched areas represent the locations where the slopes are significant at the 5% level.

To substantiate more on this, we further carried out our analysis by considered another well known heatwave index, i.e., TX90P. It is calculated as the 90th percentile of daily maximum temperatures of a five day window centred on each calendar day of a given climate reference period (1961–1990)[24]. Fig. R10 shows the relation of June TX90P with the leading mode of PMM index for the observational period of 1951–2019. In general, a positive association between PMM and TX90P is noticed over a majority of the Indian region, with the strongest signal being apparent over NCI. Thus all the three different indicators used here all points to a conclusion that the NCI region heatwaves during June are well driven the leading mode of PMM index.

(2) *There is a section (L193-L223) that shows +PMM could delay the onset of monsoon in June and suggests that the delay of the monsoon onsets can result in exceptional heatwaves over India (L197-198). I don't think the results are supportive to this argument, especially given the definition of heatwaves in this study is based on June TXx. The results or description did not explicitly connect the delayed onset of monsoon to higher TXx. intensity. I would think that a delayed onset of monsoon could extend an existing heatwave (duration matters), or increase the number of high three-day temperature (TX), or increase the probability of experiencing higher TXx (due to land-atmosphere coupling?).*

Response #5: We thank the Reviewer for this comment. Here, we would like to elaborate that the majority of moisture source during the onset period generally originates from the Indian Ocean – of which the Arabian Sea is the most dominant contributing region[25, 26, 27]. The PMM responses, on the other hand, weaken the Walker circulation and impairs the westerlies that result in reduced eastward moisture flow into the Indian mainland from the contributing major moisture sources (Fig. 3a and b). These processes have a substantial impact on the progression of monsoon rains from the north Indian Ocean towards the India mainlands, and thereby delaying the monsoon onset. Moreover, we have already shown that during the positive phase of PMM, there is substantial delay in the onset days compared to the negative of PMM – similar is the result for both mean rainy days (less rainy days during +PMM) and TXx (higher intensity during +PMM; Fig. 3c and d).

Having mention this, anecdotally, extreme heat events appear to be associated with late monsoon rains/ delayed onset monsoon, inadequate pre-monsoon rains, or low rain in neighboring regions leading to advection of dry heat into India[23]. This suggests that extreme heat waves could, in part, be a product of local heating – as suggested by the Reviewer. It is also reported that usually during the May heatwaves, clear skies led to a positive net radiation anomaly at the surface, but there is no significant sensible heat flux anomaly within the core of the heat wave affected region[28]. However, for the early June heatwaves, due to late onset of monsoon, the soil moisture drops to anomalously low levels and thereby the net surface radiation will be anomalously high, leading to a significant positive sensible heat flux anomaly developed. This leads to a substantial local forcing on air temperature that will contribute to the intensity of the heatwave event[28]. This indeed suggests that delayed onset monsoon does have an influence on the intensification of heatwaves through the land-surface processes – and thus analysing PMM influences on changes the land-surface processes can be considered as a potential future study. However, at present, we have updated these discussions in the revised manuscript.

3. The results from GFDL-FLOR need to be checked. Fig. S4 c-f, climatology from GFDL-FLOR are very off, compared to observations. It seems like the latitude is flipped. Also, in the regression plots (Fig. 2g compared to Fig. 2f), 200hPa velocity potential over Indo-WP is quite different from the observations, would this make any substantial difference in the surface responses over India?

Response #6: We thank the Reviewer for pointing this out. As they mentioned correctly, the GFDL-FLOR latitude was flipped, which is now corrected in the revised manuscript. Although we acknowledge some minor differences in the response of velocity potential over Indo-WP between observations and GFDL-FLOR control runs, the surface temperature response towards the PMM is starkly similar to what we have noticed in the observational dataset. Fig. R11a shows a positive PMM response to surface temperature is evident over the northern Indian region – specially over NCI region. Further, to emphasize more the influence of PMM on the weakening of walker circulation, we perform additional analysis us-

ing the GFDL-FLOR control runs (Fig. R11b). To this end, we analyse the relationship between strength of walker circulation (SWC) with the PMM during June. Here, the SWC is estimated by sea level pressure anomaly differences across the east and west tropical Pacific, which is associated with vertical motions of the walker circulation [13]. The described region over the tropical Pacific is 4.74°S – 4.74°N in latitude, and 128.39°E – 151.05°E and 211.47°E – 231.61°E in longitude for the western and eastern tropical Pacific edge, respectively. A significant differences in SWC are also observed between the years with strong positive (PMM⁺) and negative (PMM⁻) phases of PMM (Fig. R11b; statistically significant mean difference of -0.12 mbar based on the non-parametric bootstrap analysis; p -value < 0.05). These results are further confirms the role of PMM in weakening of walker circulation and these information is now provided in the revised manuscript (now provided as Supplementary Fig. S10 in the revised manuscript).

Figure R11: (a) Regression of June PMM index with the surface temperature considering GFDL-FLOR control runs. The black box in the plot represent the NCI region. (b) Response of strength of walker circulation (SWC) with respect to PMM; as shown in the difference between the positive and negative phase of PMM, which are represented with the box plot. The differences in SWC are statistically significant (non-parametric bootstrap analysis; p -value ≤ 0.05).

4. The authors suggested that El Nino does not have relationship (or very weak) with NCI June TXx. El Nino also has reduced OLR anomalies over the tropical central Pacific (similar to Fig. 2d), weakening the Walker Circulation. Could the authors elaborate more why El Nino has weak or no relationship with India?

Response #7: We agree with the Reviewer that the large-scale circulation response to PMM partly resembles the pattern noticed during the strong positive phase of ENSO over the Pacific, i.e. the weakening of walker circulation. Nonetheless, our analyses show a significant association between the PMM and the NCI heatwave characteristics (intensity/duration) even accounting for the confounding role of ENSO (Supplementary Fig. S8); these aspects are detailed in the manuscript and in supporting document). To further concertize the role of PMM on the NCI heatwaves – controlling other large-scale factors (e.g., ENSO) – we have now performed a climate perturbation experiment. We use atmospheric general circulation model developed by International Centre for

Theoretical Physics (ICTP AGCM)[5, 6], which could properly capture the observed pattern of climatology over India[7]. The complete description of these results and the our response to this particular comment are also provided in the Reviewer#1 comment #1.

In addition to this, the recent study by Jia et al.[29] mentioned that since 1950, a majority of ENSO events (both El Niño and La Niña) were preceded by the SST anomalies associated with the PMM mode – which is mainly initiated by a pattern of North Pacific atmospheric variability[30, 31, 32]. In conjunction to this, the model simulations also revealed a similar relationship, with a significant positive association between boreal spring PMM and ensuing winter ENSO[9, 33, 34, 35] – with even PMM events showing substantial skill in predicting the El Nino events[9, 36]. Moreover, a decadal strengthening of the PMM is also found to increase ENSO complexity[37] and it does also partly explain the shift of ENSO dynamics from eastern Pacific type to central Pacific type from around 2000[38]. Thus, it is evident from these studies that the PMM contributes to generation of ENSO and its diversity; and indicating that the PMM is an main influential trigger of tropical Pacific variability on multiple timescales[39, 40]. However, how the PMM triggering the ENSO, which affects the Indian summer climate variability is not completely understood in the literature and it is beyond the scope of the present study, and can be considered as a important future research. All these information is now provided in the revised manuscript (now provided as Supplementary Figs S7, S8 and S9 in the revised manuscript).

5. Statistic significant test should be applied to the composites shown in Fig. 1ef and regression shown in Fig.2d-i.

Response #8: We thank the Reviewer. We have now provided the statistical significant test results, where-ever it is necessary, in the revised manuscript.

6. Fig. 2d-i: the latitude and longitude ranges are different in these panels, which makes it quite difficult to compare these figures. I would suggest to use the same longitude range for every panel, and indicate the equator.

Response #9: Thank you. We appreciate the comment, but we want to keep the figure as is – which fits appropriately with our discussions in the article.

7. Fig. 4 b,c: I am not sure what those contour represent.

Response #10: The contours represents the standard deviation of the SST leading mode amongst the ensemble members. This information is now provided in the caption of the revised manuscript.

References

- [1] Lee, S.-K., Wang, C. & Mapes, B. E. A simple atmospheric model of the local and teleconnection responses to tropical heating anomalies. *Journal of Climate* **22**, 272–284 (2009).
- [2] Takaya, K. & Nakamura, H. A formulation of a phase-independent wave-activity flux for stationary and migratory quasigeostrophic eddies on a zonally varying basic flow. *Journal of the Atmospheric Sciences* **58**, 608–627 (2001).
- [3] Luo, M., Lau, N.-C., Zhang, W., Zhang, Q. & Liu, Z. Summer High Temperature Extremes over China Linked to the Pacific Meridional Mode. *J. Clim.* **33**, 5905–5917 (2020).
- [4] Ding, T., Qian, W. & Yan, Z. Changes in hot days and heat waves in China during 1961–2007. *Int. J. Climatol.* **30**, 1452–1462 (2010).
- [5] Molteni, F. Atmospheric simulations using a gcm with simplified physical parametrizations. i: Model climatology and variability in multi-decadal experiments. *Climate Dynamics* **20**, 175–191 (2003).
- [6] Kucharski, F. *et al.* On the need of intermediate complexity general circulation models: A “speedy” example. *Bulletin of the American Meteorological Society* **94**, 25–30 (2013).
- [7] Hari, V., Pathak, A. & Koppa, A. Dual response of arabian sea cyclones and strength of indian monsoon to southern atlantic ocean. *Climate Dynamics* **56**, 2149–2161 (2021).
- [8] Thomas, E. E. & Vimont, D. J. Modeling the Mechanisms of Linear and Non-linear ENSO Responses to the Pacific Meridional Mode. *J. Clim.* **29**, 8745–8761 (2016).
- [9] Larson, S. M. & Kirtman, B. P. The Pacific Meridional Mode as an ENSO Precursor and Predictor in the North American Multimodel Ensemble. *J. Clim.* **27**, 7018–7032 (2014).
- [10] Di Lorenzo, E. *et al.* ENSO and meridional modes: A null hypothesis for Pacific climate variability. *Geophys. Res. Lett.* **42**, 9440–9448 (2015).
- [11] Liu, Z., Gao, T., Zhang, W. & Luo, M. Implications of the Pacific meridional mode for summer precipitation extremes over China. *Weather Clim. Extremes* **33**, 100359 (2021).
- [12] Zhang, W., Vecchi, G. A., Murakami, H., Villarini, G. & Jia, L. The Pacific Meridional Mode and the Occurrence of Tropical Cyclones in the Western North Pacific. *J. Clim.* **29**, 381–398 (2016).
- [13] Zhao, X. & Allen, R. J. Strengthening of the Walker Circulation in recent decades and the role of natural sea surface temperature variability. *Environ. Res. Commun.* **1**, 021003 (2019).
- [14] Amaya, D. J. *et al.* The north pacific pacemaker effect on historical enso and its mechanisms. *Journal of Climate* **32**, 7643–7661 (2019).

- [15] Suarez-Gutierrez, L., Milinski, S. & Maher, N. Exploiting large ensembles for a better yet simpler climate model evaluation. *Clim. Dyn.* **57**, 2557–2580 (2021).
- [16] Tokarska, K. B. *et al.* Past warming trend constrains future warming in cmip6 models. *Science advances* **6**, eaaz9549 (2020).
- [17] Huang, X. *et al.* South Asian summer monsoon projections constrained by the interdecadal Pacific oscillation. *Sci. Adv.* (2020). URL <https://www.science.org/doi/10.1126/sciadv.aay6546>.
- [18] Gill, E. C., Rajagopalan, B. & Molnar, P. Subseasonal variations in spatial signatures of ENSO on the Indian summer monsoon from 1901 to 2009. *J. Geophys. Res. Atmos.* **120**, 8165–8185 (2015).
- [19] Field, C. B., Barros, V., Stocker, T. F. & Dahe, Q. *Managing the risks of extreme events and disasters to advance climate change adaptation: special report of the intergovernmental panel on climate change* (Cambridge University Press, 2012).
- [20] García-León, D. *et al.* Current and projected regional economic impacts of heatwaves in Europe - Nature Communications. *Nat. Commun.* **12**, 1–10 (2021).
- [21] Perkins, S. E. & Alexander, L. V. On the measurement of heat waves. *Journal of climate* **26**, 4500–4517 (2013).
- [22] Meehl, G. A. & Tebaldi, C. More intense, more frequent, and longer lasting heat waves in the 21st century. *Science* **305**, 994–997 (2004).
- [23] Ratnam, J. V., Behera, S. K., Ratna, S. B., Rajeevan, M. & Yamagata, T. Anatomy of Indian heatwaves - Scientific Reports. *Sci. Rep.* **6**, 1–11 (2016).
- [24] Donat, M. *et al.* Updated analyses of temperature and precipitation extreme indices since the beginning of the twentieth century: The hadex2 dataset. *Journal of Geophysical Research: Atmospheres* **118**, 2098–2118 (2013).
- [25] Roxy, M. K. *et al.* A threefold rise in widespread extreme rain events over central India. *Nat. Commun.* **8**, 1–11 (2017).
- [26] Pathak, A., Ghosh, S. & Kumar, P. Precipitation recycling in the indian subcontinent during summer monsoon. *Journal of Hydrometeorology* **15**, 2050–2066 (2014).
- [27] Pathak, A., Ghosh, S., Martinez, J. A., Dominguez, F. & Kumar, P. Role of oceanic and land moisture sources and transport in the seasonal and interannual variability of summer monsoon in india. *Journal of Climate* **30**, 1839–1859 (2017).
- [28] Ghatak, D., Zaitchik, B., Hain, C. & Anderson, M. The role of local heating in the 2015 Indian Heat Wave - Scientific Reports. *Sci. Rep.* **7**, 1–8 (2017).
- [29] Jia, F., Cai, W., Gan, B., Wu, L. & Di Lorenzo, E. Enhanced North Pacific impact on El Niño/Southern Oscillation under greenhouse warming - Nature Climate Change. *Nat. Clim. Change* **11**, 840–847 (2021).

- [30] Chiang, J. C. H. & Vimont, D. J. Analogous Pacific and Atlantic Meridional Modes of Tropical Atmosphere–Ocean Variability. *J. Clim.* **17**, 4143–4158 (2004).
- [31] Vimont, D. J., Alexander, M. & Fontaine, A. Midlatitude Excitation of Tropical Variability in the Pacific: The Role of Thermodynamic Coupling and Seasonality. *J. Clim.* **22**, 518–534 (2009).
- [32] Yu, J.-Y. & Kim, S. T. Relationships between Extratropical Sea Level Pressure Variations and the Central Pacific and Eastern Pacific Types of ENSO. *J. Clim.* **24**, 708–720 (2011).
- [33] Zhang, L., Chang, P. & Ji, L. Linking the Pacific Meridional Mode to ENSO: Coupled Model Analysis. *J. Clim.* **22**, 3488–3505 (2009).
- [34] Lin, C.-Y., Yu, J.-Y. & Hsu, H.-H. CMIP5 model simulations of the Pacific meridional mode and its connection to the two types of ENSO. *Int. J. Climatol.* **35**, 2352–2358 (2015).
- [35] Ma, J., Xie, S.-P. & Xu, H. Contributions of the North Pacific Meridional Mode to Ensemble Spread of ENSO Prediction. *J. Clim.* **30**, 9167–9181 (2017).
- [36] Amaya, D. J. The Pacific meridional mode and ENSO: A review. *Current Climate Change Reports* **5**, 296–307 (2019).
- [37] Yu, J.-Y. & Fang, S.-W. The Distinct Contributions of the Seasonal Footprinting and Charged-Discharged Mechanisms to ENSO Complexity. *Geophys. Res. Lett.* **45**, 6611–6618 (2018).
- [38] Yu, J.-Y., Kao, H.-Y. & Lee, T. Subtropics-Related Interannual Sea Surface Temperature Variability in the Central Equatorial Pacific. *J. Clim.* **23**, 2869–2884 (2010).
- [39] Min, Q., Su, J., Zhang, R. & Rong, X. What hindered the El Niño pattern in 2014? *Geophys. Res. Lett.* **42**, 6762–6770 (2015).
- [40] You, Y. & Furtado, J. C. The South Pacific Meridional Mode and Its Role in Tropical Pacific Climate Variability. *J. Clim.* **31**, 10141–10163 (2018).

REVIEWER COMMENTS

Reviewer #1 (Remarks to the Author):

Thank authors for their comprehensive responses to most of my concerns, especially on the aspect of interannual variability. They implemented AGCM experiments and analyzed large ensemble runs to show more robust information on the interannual relation between PMM and NCI heatwaves. However, there still exist unclear arguments in the response which does not support this study to be published.

1. For the comment #4, authors responded that "We agree with the Reviewer that a prominent low-level cyclone is seen over the western North Pacific ...". But my question was about the cyclonic over the core PMM region. Why did you talk about the anomalous cyclone over the western North Pacific? The core PMM region is defined in the domain (175E-95W and 21S-32N) in Fig. 1f. However, main part of the cyclonic anomaly (30-45N) in Fig. 2d is out of the core PMM region. To confirm whether the cyclone can respond to PMM SST anomalies, authors could show the results from their AGCM experiment (Fig R2a).

2. For the comment #8, authors show the result from MIROC6. It rightly confirms my worry that that internal variability could make the chosen "adequate models" unreliable. As shown in Fig. R6a, the relation between NCI and PMM in this single model MIROC6 can spread from -0.15 to 0.35, very unstable. If you analyze the member with low correlation -0.15, you think MIROC6 is not a good model to be used to constrain projection. But if you use a high correlation member you may think MIROC6 is an adequate model. To judge if a model belongs to the "adequate models" depend on the member chosen. It should be the same for other models. Very small members of the CMIP6 models used in this study (Table S1) would mislead the constrained result.

Reviewer #2 (Remarks to the Author):

The manuscript has been largely improved and the new title is indeed a much better fit. I appreciated the authors' efforts in responding all my comments carefully. Based on the results in this manuscript, I've found it is convincing that the Pacific Meridional Mode (PMM) has robust impacts on the onset of Indian summer monsoon and the NC Indian summer temperature variability. I have a few comments, hopefully they can help to improve the manuscript further. Also, there are many minor issues and lack of information throughout the manuscript.

1. For the last part of the results (Implications of adequate representation of PMM in the CMIP6 models), I understand that this part is served as a secondary information, but I have trouble to get the main message from this part. I could interpret in different ways: (1) the relationship between +PMM and +TXx over the NCI would be stronger in the future; (2) among different factors that can affect the NCI TXx, the relative contribution from PMM would become larger in the future; (3) to estimate the degree of warming TXx over the NCI in the future, CMIP6 models project a wide range of warming, but among them, the models that have better performance in simulating PMM-NCI TXx relationship simulate warmer NCI projection.
Even serving as secondary part, I think it is still necessary to convey the main purpose of this part clearly.
2. L462 & L470 “the PMM response to the NCI heatwave intensities”: I think it should be the other way around? The PMM affects the NCI heatwave intensities, so it should be the NCI heatwave intensities response to the PMM. Also, L470 doesn't explicitly mention that $\overline{\Delta TX_x}$ is the differences between the composites of TX_x during +PMM minus the ones during -PMM (as indicated in Table S1).
3. Fig. S13 & L320-323: (1) I think it is necessary to show surface wind and OLR anomalies that include the Indian Ocean and India (like Fig. 2e). Fig. S13 only shows the circulations anomalies over the tropical Pacific, which is not enough to demonstrate how change in large-scale circulations affect the warming over India. (2) Was the velocity potential shown in Fig. S13 detrended? If not, is it possible that the results here are just the projected warming in the basic-state, not necessary demonstrating the relationship between the PMM and TXx over India?
4. I understand that the PMM is a seasonal phenomenon, so using May or June data would not make much differences. However, as the entire argument is based on how PMM remote teleconnections affect “early June” circulations and moisture transport over the Indian Ocean and India, I would think the anomalous large-scale circulations have been set up in May so it could affect the anomalies over the NCI in early June. Thus, wouldn't it be more reasonable to show the PMM-related large-scale circulations in May? Otherwise, June data includes a large portion of dates that happen after the onset of monsoon.

5. L215-233: I would suggest the authors to elaborate more on the results from GFDL-FLOR control runs, especially the circulations and OLR anomalies over the Indian Ocean and India (Fig. 2e). I think it is clear that, in the control simulations, +PMM teleconnections imposes significant enhanced OLR over northern India. This is a quite key result.
6. How did the authors define ENSO events? And what is the criterion of strong ENSO?

Minor comments:

L23 “emerging signatures of PMM”: what does emerging signature mean here? Emerging signal is often used as when anthropogenic forcing emerges. I don’t think this manuscript has touched any of this. So please consider rephrasing to avoid misunderstanding.

L97-101: I understand that Fig. 1a served as motivation only, but this statement is largely inaccurate. The most fatal heatwaves in late May to early June shown in Fig. 1a are not consistent with the events in June shown in Fig. 1d (e.g., among the top 5 fatal heatwaves, only 2003 event is also listed in Fig. 1d).

L147: “equatorial” eastern Pacific

L149: central-eastern tropical Pacific is more accurate.

L154 SWC: the acronym is unnecessary.

L159-160 Matsuno-Gill response: I can’t tell the Matsuno-Gill response. Also, isn’t the latitude too far from the tropics?

L164-184: I can’t tell the relevance of this paragraph to the context.

L185-187: Is this possibly due to the fact that PMM is a precursor of ENSO? And some ENSO events have developed in late spring?

Fig. S7: the ENSO index used here, is it June monthly data? Or 3-month averaged, a common way to define ENSO, data?

L194 “concertize”: Merriam-Webster define this word as “to perform professionally in concerts”, it doesn’t fit into the context.

Supplementary note 4: (1) please fix the figure numbers in the note. (2) what exactly does seasonal climatology mean here? Monthly climatology or three-month averaged climatology, and if it’s the latter, are they overlapping? (3) Based on the Fig. S9, only “regional” SST anomalies were prescribed. This should be explicit in the note 4.

L233: or model errors or/and biases.

L234 Response of PMM to moisture transport and monsoon onset:

Should it be the other way around? The responses of moisture transport and monsoon onset to the PMM because it is the PMM imposes impacts and moisture transport & monsoon onset respond to the impacts imposed by the PMM.

L259 “mean differences”: differences between +PMM and -PMM?

L260-262: I would suggest to provide some context. “local heating” seems to be out of sudden.

Fig S12: responses -> regression or composites?

L295-296: what do $N=6$ and ρ mean respectively here?

Table S1: please indicate the models that are categorized as “adequate”.

Fig. S13: please show the region west to 100E, consistent with Fig. 2.

L397: what is the horizontal resolution for HadISST?

L450-455: the processes to define PMM should be mentioned in L383 paragraph. In L383 paragraph, the authors didn't mention anything about detrend, 3-month average on SST ...etc these critical information.

Reviewer #1:

Thank authors for their comprehensive responses to most of my concerns, especially on the aspect of interannual variability. They implemented AGCM experiments and analyzed large ensemble runs to show more robust information on the interannual relation between PMM and NCI heatwaves. However, there still exist unclear arguments in the response which does not support this study to be published.

Response #1: We sincerely thank the Reviewer for their constructive comments – which have certainly helped us to improve our manuscript further. Now, in this revision we have addressed all your comments and we believe that our manuscript has improved in its focus and clarity.

1. For the comment #4, authors responded that "We agree with the Reviewer that a prominent low-level cyclone is seen over the western North Pacific ...". But my question was about the cyclonic over the core PMM region. Why did you talk about the anomalous cyclone over the western North Pacific? The core PMM region is defined in the domain (175E–95W and 21S–32N) in Fig. 1f. However, main part of the cyclonic anomaly (30–45N) in Fig. 2d is out of the core PMM region. To confirm whether the cyclone can respond to PMM SST anomalies, authors could show the results from their AGCM experiment (Fig R2a).

Response #2: We are sorry that we have overlooked your earlier comment in this respect. We agree with the Reviewer at this point that the anomalous cyclone occurred in the region between 30–45N. As suggested by the reviewer, we now show the results based on the AGCM experiment to ascertain and confirm whether the cyclone can respond to PMM SST anomalies. In the below Fig. R1 based on analysis of the AGCM experiment indeed shows an anomalous cyclonic circulation between the 30–45N region to the response of PMM. We also note here that similar observations have been made in an earlier study by Zhang et al.[1], where they implemented a high-resolution Geophysical Fluid Dynamics Laboratory (GFDL) Forecast-Oriented Low Ocean Resolution (FLOR) coupled climate model.

2. For the comment #8, authors show the result from MIROC6. It rightly confirms my worry that that internal variability could make the chosen "adequate models" unreliable. As shown in Fig. R6a, the relation between NCI and PMM in this single model MIROC6 can spread from -0.15 to 0.35, very unstable. If you analyze the member with low correlation -0.15, you think MIROC6 is not a good model to be used to constrain projection. But if you use a high correlation member you may think MIROC6 is an adequate model. To judge if a model belongs to the "adequate models" depend on the member chosen. It should be the same for other models. Very small members of the CMIP6 models used in this study (Table S1) would mislead the constrained result.

Response #3: We thank the reviewer for this insightful comment. This really brings us back to the working table as to better and clearly convey the main message of our work/analysis. We agree with the reviewer observation that the MIROC6 based simulations had a large spread in their correlation estimates ranging from -0.15 to 0.35. As noted earlier and

Figure R1: Response of 850 winds and outgoing longwave radiation (OLR) to the summer PMM perturbation over the Pacific ocean from the AGCM experiment.

also provided in the supplementary document – more than 66% of the ensembles exhibited a positive association with the PMM index, specifically over the NCI region (Supplementary Fig. S12). Thus conveying the degree of consensus between model ensembles in the physical response to summer PMM forcing and this number can be considered adequate in communicating the uncertainty based on climate model ensembles as mentioned in the previous studies[2, 3] – and even in the IPCC Summary for Policymakers[4]. To this end, we also would like to note a similar procedure adopted in the recent study by Huang et al. [5] – focusing on the projections of Indian summer monsoon rainfall – wherein the authors identified that the performance of large ensemble (internal variability) in reproducing the climatology and historical changes of Indian monsoon in association with interdecadal Pacific oscillation aid confidence – as some of the ensemble members (more than 66%, same as ours) can reproduce the observed changes.

Having mentioned this, we understand the concern of the Reviewer – even with all these detailed analyses, we believe the “adequate model” is not a right choice of wording in the context of our study. Indeed such a choice of wording has masked the main findings of this study; and have put more emphasis on results for the projections based on climate model simulations. We emphasize that the main finding of our study is the far-reaching impact of PMM and its novel association to the extreme temperatures over the northern Indian region. Along with their overlying large-scale atmospheric circulation pattern, we demonstrate these associations through both observations and climate model based control and perturbation experiment analyses. As a secondary piece of information, we then proceed with showing the usefulness and implication of this novel association for the extreme temperature projections from CMIP6 climate model projections. To this end the idea was to see if we can see the coherent and consistent differences in future temperature projections be-

tween different group/ensemble of climate model simulations. We indeed find and report that a set of models (ensembles) that better capture the observed PMM response, have significantly different (higher) projected changes in temperature extremes over India as compared to the rest of the models/ensembles. As to why such differences lie between two set of models/ensembles – we hint this to their abilities to capture observed PMM response – and certainly this would require a more detail and comprehensive study to further investigate looking for example underlying individual model physically mechanisms and/or tracking a trajectory of particular realisation. We ask here for the understanding of the reviewer that certainly such a study in-itself an interesting one would certainly be beyond the scope of this (limited format) communication work. We have now exclusively added these arguments in our revised manuscript (Lines 325–341; Page No. 18–19).

What is more compelling from our (observed PMM response) analysis for the future temperature projections is that not only that the grouping of different climate models, but also grouping of a single model with different realisations – both of these analyses pointed out a similar aspect that ones that adequately capture the observed PMM response projected higher temperature in a consistent way. To further accretion this aspect, we further conducted an additional analysis, however, this time using the MPI-GE ensembles with 100 realizations[6]. Similar to that of MIROC6, in evaluating the historical PMM index and NCI temperature, we found that all the MPI-GE ensemble exhibited a positive relationship between summer PMM and NCI temperature (Supplementary Fig. S12 and Fig. R2a). Now separating these ensembles in two groups, a set of realisations with the higher positive association with the PMM index – similar or above the observed value – and the rest ones (separated by the red and blue circles in Fig. R2a). We then analyse the changes in the future temperature projections based on these two groups. A clear and significant increase in the temperature changes (approximately 0.10°C) are projected by the ensembles which relatively better captures the observed PMM response – compared to the rest of the ensembles (Fig. R2b). This is consistent and in-line with our broader findings for the projection part (CMIP6 simulations and MIROC6 based ensemble realisations).

In summary, we understand that naming the models as “adequate” and “inadequate” has created some confusions – and as such implies about the validity of some climate model simulations. This part of overall questioning and querying about the projections made by CMIP6 simulations was certainly not the main aim of the study. Rather these projection-related analyses were done to show the usefulness/implications of the so-identified novel PMM association and their responses to projected temperature extremes over the NCI region. As we see these naming issues as a source of confusion, we have now renamed the models/realizations from a more neutral tone, namely as: “Set A” (which captures the observed pattern of PMM and NCI associations) and “Set B” (which don’t capture). We highlighted these issues in the revised manuscript. We would like to note here that based on the earlier suggestion of Reviewer #2,

we have changed the title, which now shed more light/importance on the observed PMM response and subsequent association part than the future projection ones. We hope that the reviewer can now more clearly see and appreciate our views regarding the analysis and findings of this study.

Figure R2: **Evaluation of the historical PMM and NCI temperature in large ensemble** (a) The NCI temperature relation with the summer PMM index for the observed (black cross mark) and the MPI members (blue circles) during 1951-2014. The members with the higher positive association with the PMM index – which is similar or above the observed value – are shown in red circles. (b) Box-plots depicting the projected changes in the surface temperature intensities under RCP8.5 scenario over NCI region during the second half of the twenty-first century (2065–2100) w.r.t the historical simulations (1980–2014) for the members with the higher positive association with the PMM (red colored) and for the rest of the members (blue colored).

Reviewer #2:

The manuscript has been largely improved and the new title is indeed a much better fit. I appreciated the authors' efforts in responding all my comments carefully. Based on the results in this manuscript, I've found it is convincing that the Pacific Meridional Mode (PMM) has robust impacts on the onset of Indian summer monsoon and the NC Indian summer temperature variability. I have a few comments, hopefully they can help to improve the manuscript further. Also, there are many minor issues and lack of information throughout the manuscript.

Response #1: We thank the Reviewer for appreciating our efforts in our previous revised manuscript. We have now addressed all their remaining comments/suggestions in this revised version – which have certainly improved the clarity of the presented work.

1. For the last part of the results (Implications of adequate representation of PMM in the CMIP6 models), I understand that this part is served as a secondary information, but I have trouble to get the main message from this part. I could interpret in different ways: (1) the relationship between +PMM and +TX_x over the NCI would be stronger in the future; (2) among different factors that can affect the NCI TX_x,

the relative contribution from PMM would become larger in the future; (3) to estimate the degree of warming TXx over the NCI in the future, CMIP6 models project a wide range of warming, but among them, the models that have better performance in simulating PMM-NCI TXx relationship simulate warmer NCI projection. Even serving as secondary part, I think it is still necessary to convey the main purpose of this part clearly.

Response #2: We are thankful for the Reviewer for pointing this out. We emphasize that the main finding of our study is the far-reaching impact of PMM and its novel association to the extreme temperatures over the northern Indian region. Along with their overlying large-scale atmospheric circulation pattern, we demonstrate these associations through both observations and climate model based control and perturbation experiment analyses. As a secondary piece of information, we then proceed with showing the usefulness and implication of this novel association for the extreme temperature projections from CMIP6 climate model projections. As indicated by the Reviewer himself that the CMIP6 models project a wide range of warming, but among them, the models that have better performance in simulating PMM-NCI TXx relationship in the historical period simulate warmer NCI projection. Accordingly we have now conveyed/highlighted this aspect in our revised manuscript (Lines 325–341; Page No. 18–19).

2. L462 & L470 "the PMM response to the NCI heatwave intensities": I think it should be the other way around? The PMM affects the NCI heatwave intensities, so it should be the NCI heatwave intensities response to the PMM. Also, L470 doesn't explicitly mention ΔTX_X is the differences between the composites of during +PMM minus the ones during -PMM (as indicated in Table S1).

Response #3: We are thankful to the Reviewer for bringing these points to our notice. The reviewer is correct in their remark. Accordingly, we have now modified the texts (Lines 465 and 474; Page No. 26).

3. Fig. S13 & L320-323: (1) I think it is necessary to show surface wind and OLR anomalies that include the Indian Ocean and India (like Fig. 2e). Fig. S13 only shows the circulations anomalies over the tropical Pacific, which is not enough to demonstrate how change in large-scale circulations affect the warming over India. (2) Was the velocity potential shown in Fig. S13 detrended? If not, is it possible that the results here are just the projected warming in the basic-state, not necessary demonstrating the relationship between the PMM and TXx over India?

Response #4: We thank the Reviewer for this comment. As per their suggestion, we have now modified this particular figure in the revised manuscript by: (i) adding the OLR and wind responses to PMM over the Indian Ocean and Indian region, and (ii) de-trending all the variables including OLR, winds and velocity potential. Overall with all these changes, the results are consistent with those reported in earlier version.

4. I understand that the PMM is a seasonal phenomenon, so using May or June data would not make much differences. However, as the entire argument is based on how PMM remote teleconnections affect "early June" circulations and moisture transport over the Indian Ocean and India, I would think the anomalous large-scale

circulations have been set up in May so it could affect the anomalies over the NCI in early June. Thus, wouldn't it be more reasonable to show the PMM-related large-scale circulations in May? Otherwise, June data includes a large portion of dates that happen after the onset of monsoon.

Response #5: We thank the Reviewer for this remark, and we agree with their concern. Considering your suggestion, we have now analysed the PMM-related large-scale circulations in May. Fig. R3a shows a weakening of westerlies along with an increased OLR over NCI – similar to those noticed for the June month (Fig. 2 in the main manuscript). These responses are accompanied with the increased divergence (convergence) over the eastern Pacific at the 200 (850) hpa pressure level (Fig. R3b). Over the western Pacific region, however, the convergence at the higher level (200 hpa) is noticed, resulting in a reduced convection over the region. These convergence/divergence changes in easterlies/westerlies signify the weakening of a Walker circulation pattern. Similar observations were made when we performed the analysis considering the May and June months together as well (Fig. R3c and d). This analysis indicates that the anomalous large-scale circulation have been set up in May so it could affect the anomalies over the NCI in early June – as also remarked by the Reviewer. We have added these in the revised manuscript (See supplementary Fig. S15 in the revised manuscript).

5. L215-233: I would suggest the authors to elaborate more on the results from GFDL-FLOR control runs, especially the circulations and OLR anomalies over the Indian Ocean and India (Fig. 2e). I think it is clear that, in the control simulations, +PMM teleconnections imposes significant enhanced OLR over northern India. This is a quite key result.

Response #6: We thank the Reviewer and agree with them in this regard. As per their suggestion, we have now added a sentence "Moreover, the substantial increase in an OLR is noted over the Indian region, indicating PMM teleconnections imposes significant heatwave conditions over NCI." in the revised manuscript (Lines 211–213; Page No. 12).

6. How did the authors define ENSO events? And what is the criterion of strong ENSO?

Response #7: El Niño and La Niña events are defined as the Niño-3.4 index in five consecutive overlapping 3-month periods above 0.5°C and below -0.5°C , respectively (following https://origin.cpc.ncep.noaa.gov/products/analysis_monitoring/ensostuff/ONI_v5.php). This information is provided in the manuscript (see Supplementary Fig. S8).

Minor comments:

L23 "emerging signatures of PMM": what does emerging signature mean here? Emerging signal is often used as when anthropogenic forcing emerges. I don't think this manuscript has touched any of this. So please consider rephrasing to avoid misunderstanding.

Response #8: We understand the concern of the Reviewer and accord-

Figure R3: (a) Large-scale atmospheric circulation pattern associated with the PMM during May in terms of the regression of outgoing long-wave radiation (OLR; shading during 1975–2019) and winds at 850 hPa (vectors, during 1951–2019) from NCEP reanalysis. The response of velocity potential at 200 (b; upper panel) and 850 (b; lower panel) hPa levels to the leading PMM index, wherein the negative (positive) values indicates the region with increased divergence (convergence). (c) and (d) is same as (a) and (b), but for the months May and June.

ingly modified the related texts in the revised manuscript.

L97-101: I understand that Fig. 1a served as motivation only, but this statement is largely inaccurate. The most fatal heatwaves in late May to early June shown in Fig. 1a are not consistent with the events in June shown in Fig. 1d (e.g., among the top 5 fatal heatwaves, only 2003 event is also listed in Fig. 1d).

Response #9: We understand the concern of the Reviewer. However, with all due respect we point here that the Fig. 1a shows the mortality due to heatwaves for all of India – and this information is used as a motivation to analyse/understand large-scale drivers of heatwaves in the pre-monsoon period. But the Fig. 1d is specifically for the NCI region. This is now mentioned clearly in the revised manuscript (Lines 99–100; Page No. 6).

L147: “equatorial” eastern Pacific

Response #10: Done

L149: central-eastern tropical Pacific is more accurate.

Response #11: Thanks and modified accordingly.

L154 SWC: the acronym is unnecessary.

Response #12: Thanks and modified accordingly.

L159-160 Matsuno-Gill response: I can't tell the Matsuno-Gill response. Also, isn't the latitude too far from the tropics?

Response #13: We agree with the Reviewer. We have removed this specifics in the revised manuscript.

L164-184: I can't tell the relevance of this paragraph to the context.

Response #14: We understand the concern of the Reviewer. We provided this paragraph to show that PMM not only influences the weakening of walker circulation but also induces an eastward propagation of Rossby waves to explain the wave activities between 30N–60N as shown in Supplementary Fig. S1 – which was also one of the comments raised by the Reviewer #1 during our previous submission. However, considering the Reviewer #2 suggestion, we also feel that these information are probably obsolete (or not so necessary). Accordingly, we have removed this paragraph and added only a short note to this context in the revised manuscript (Lines 162–168; Page No. 9–10).

L185-187: Is this possibly due to the fact that PMM is a precursor of ENSO? And some ENSO events have developed in late spring?

Response #15: Thank you for this important remark. Some studies have reported that PMM is coupled with ENSO through a wind–evaporation–SST feedback mechanism[7], sometime even acting as a precursor and trigger of ENSO[8]. However, to confidently say anything on this related to our analysis would require a separate detailed study – which is beyond the scope of the current work. We nevertheless did made a note of this aspect in the manuscript on the possible role of PMM on ENSO (Lines 192–195; Page No. 11).

Fig. S7: the ENSO index used here, is it June monthly data? Or 3-month averaged, a common way to define ENSO, data?

Response #16: For this figure we have used the June ENSO index. However, the 3-month averaged – a more common way of defining the ENSO index – is also considered and analysis of this is presented in Supplementary Figure S8. We have now mentioned this specifically in the figure caption in the revised manuscript.

L194 “concertize”: Merriam-Webster define this word as “to perform professionally in concerts” , it doesn't fit into the context.

Response #17: Thanks, we have rephrased this word in the revised manuscript.

Supplementary note 4: (1) please fix the figure numbers in the note. (2) what exactly does seasonal climatology mean here? Monthly climatology or three-month averaged climatology, and if it's the latter, are they overlapping? (3) Based on the Fig. S9, only “regional” SST anomalies were prescribed. This should be explicit in the note

4.

Response #18: (i) the Figure numbers in the note are now fixed in the revised manuscript. (ii) it is monthly climatology and it is now clearly mentioned in the revised Supplementary note. (iii) Yes, only regional SST anomalies were perturbed both for PMM and ENSO region, and it is now explicitly mentioned in the Supplementary note 4.

L233: or model errors or/and biases

Response #19: Thanks, we have added those in the revised manuscript.

L234 Response of PMM to moisture transport and monsoon onset: Should it be the other way around? The responses of moisture transport and monsoon onset to the PMM because it is the PMM imposes impacts and moisture transport & monsoon onset respond to the impacts imposed by the PMM.

Response #20: Thank you for these remarks. We agree with the Reviewer and have accordingly modified the related texts in the revised manuscript.

L259 “mean differences”: differences between +PMM and -PMM?

Response #21: Yes, this information is now explicitly mentioned in the manuscript, ”a profound difference in the rainy days are observed between the years with strong positive and negative phases of the PMM”. See Lines 243–244 and Page No. 14

L260-262: I would suggest to provide some context. “local heating” seems to be out of sudden.

Response #22: We understand the concern of the Reviewer and have now provided the context in the revised manuscript (Lines 248–250; Page No. 14).

Fig S12: responses:: regression or composites?

Response #23: It is regression – which is now mentioned in the revised manuscript.

L295-296: what do $N=6$ and ρ mean respectively here?

Response #24: N here refers to the number of models which adequately represent the PMM and NCI heatwave relation; and ρ is the correlation coefficient between the PMM SST expansion coefficients of climate model and observations. Mean value refers to the average of six ρ values.

Table S1: please indicate the models that are categorized as “adequate”.

Response #25: Done

Fig. S13: please show the region west to 100E, consistent with Fig. 2.

Response #26: Done

L397: what is the horizontal resolution for HadISST?

Response #27: The horizontal resolution is one degree – which is now mentioned in the revised manuscript.

L450-455: the processes to define PMM should be mentioned in L383 paragraph. In L383 paragraph, the authors didn't mention anything about detrend, 3-month average on SST ... etc these critical information

Response #28: Thank you. We have now provided these details in the Method part of the revised manuscript (Lines 386–388; Page No. 21–22.

References

- [1] Zhang, W., Vecchi, G. A., Murakami, H., Villarini, G. & Jia, L. The Pacific Meridional Mode and the Occurrence of Tropical Cyclones in the Western North Pacific. *J. Clim.* **29**, 381–398 (2016).
- [2] McInnes, K. L., Erwin, T. A. & Bathols, J. M. Global climate model projected changes in 10 m wind speed and direction due to anthropogenic climate change. *Atmospheric Science Letters* **12**, 325–333 (2011).
- [3] Power, S. B., Delage, F., Colman, R. & Moise, A. Consensus on twenty-first-century rainfall projections in climate models more widespread than previously thought. *Journal of Climate* **25**, 3792–3809 (2012).
- [4] Alley, R. B. & Coauthors. Summary for policymakers. climate change 2007: The physical science basis. In *Contribution of Working Group I to the fourth assessment report of the Intergovernmental Panel on Climate Change*, 1–13 (Cambridge University Press Cambridge, UK, 2007).
- [5] Huang, X. *et al.* South asian summer monsoon projections constrained by the interdecadal pacific oscillation. *Science advances* **6**, eaay6546 (2020).
- [6] Maher, N. *et al.* The max planck institute grand ensemble: enabling the exploration of climate system variability. *Journal of Advances in Modeling Earth Systems* **11**, 2050–2069 (2019).
- [7] Thomas, E. E. & Vimont, D. J. Modeling the mechanisms of linear and nonlinear ENSO responses to the Pacific meridional mode. *Journal of Climate* **29**, 8745–8761 (2016).
- [8] Larson, S. M. & Kirtman, B. P. The Pacific meridional mode as an ENSO precursor and predictor in the North American multimodel ensemble. *Journal of Climate* **27**, 7018–7032 (2014).

REVIEWERS' COMMENTS

Reviewer #1 (Remarks to the Author):

Authors have fully responded to my concerns. I have no more comments.

Reviewer #2 (Remarks to the Author):

Reviewer comments: NCOMMS-21-12439A "Strong influence of north Pacific Ocean variability on Indian summer heatwaves"

I would recommend the manuscript is ready to be published. I only have some minor comments:

L25-26: Consider rephrasing. "These differences" is vague, not sure what the "differences" are.

L165-168: I cannot really tell the similarity between Figs.S1 and S6. Also, please replot Fig.S6, centering on the Pacific, as the focus here is a wave train propagating across the North Pacific, it doesn't make sense to center on the Atlantic and split the Pacific. It's also very difficult to compare with Fig.S1.

TableS1: what is the observed value for Txx_PMM+ minus Txx_PMM-

L356-358: Consider rephrasing. Not sure what "climatological bias" means in this context.

ICTP AGCM in Supplementary Note4: what are the years of HadISST used for the climatology

L387: Was 3-month running before or after removing the seasonal cycle and detrending the data?

L411 & 415: European drought?

L503-511: Consider to re-polish this paragraph. I've found it is not easy to connect this part to understand how the statistical significance in L302-305 (Fig.4b) was conducted (the differences between Txx in the Set A and Txx in all models)

Fig.S13 e,f: Try not to saturate the colorbar, it is quite difficult to read and compare.

Reviewer #1:

Authors have fully responded to my concerns. I have no more comments.

Response #1: We thank the reviewer for their time and efforts in reviewing our work.

Reviewer #2:

I would recommend the manuscript is ready to be published. I only have some minor comments.

Response #1: We thank the Reviewer for appreciating our efforts in our previous revision. We have now addressed all their remaining comments/suggestions in this revised version – which have certainly improved the clarity of the presented work.

L25-26: Consider rephrasing. “These differences” is vague, not sure what the “differences” are.

Response #2: We agree and accordingly have modified this sentence in the revised manuscript (see Line 25).

L165-168: I cannot really tell the similarity between Figs.S1 and S6. Also, please replot Fig.S6, centering on the Pacific, as the focus here is a wave train propagating across the North Pacific, it doesn't make sense to center on the Atlantic and split the Pacific. It's also very difficult to compare with Fig.S1.

Response #3: We partly agree with the Reviewer in this regard. The intention of this figure/analysis (Fig. S6) is to provide the information pertaining to the PMM influence on the wave train propagation – the similarity of the spatial pattern of which is also to a certain degree being noticed in Fig. S1. Such eastward propagation is also reported in previous studies – especially during the summer months[1, 2]. This though partial resemblance of the wave train patterns are noticed in both the figures S1 and S6, which we have now mentioned in the revised manuscript. Also, following the Reviewer suggestion, we have now modified the figure S6, centering the plot on the Pacific region. To further increase the readability of this supplemental Fig. S6, we now just provided the geopotential and wind patterns and removed the stream-function plots as the latter is just a supplement to the geopotential height.

Table S1: what is the observed value for Txx_{PMM+} minus Txx_{PMM-}

Response #4: We have now added this information in the Table S1 of the revised manuscript.

L356-358: Consider rephrasing. Not sure what “climatological bias” means in this context.

Response #5: We have now modified the “climatological bias” to “sub-

stantial bias” in the revised manuscript.

ICTP AGCM in Supplementary Note4: what are the years of HadISST used for the climatology.

Response #6: This information is now provided in the revised manuscript.

L387: Was 3-month running before or after removing the seasonal cycle and detrending the data?

Response #7: It is applied after removing seasonal cycle and detrending the data. This information is explicitly mentioned in the revised manuscript.

L411 & 415: European drought?

Response #8: This is an inadvertent error which is now corrected in the revised manuscript.

L503-511: Consider to re-polish this paragraph. I've found it is not easy to connect this part to understand how the statistical significance in L302-305 (Fig.4b) was conducted (the differences between Txx in the Set A and Txx in all models)

Response #9: Thank you for your remark. As elaborated in the methodology we have tested the significant by re-sampling all CMIP6 model ensembles for 10 000 times with a sample size similar to the Set A model (varying between five and seven depending on the threshold criterion to classify adequate models). For each realization, we then estimate the respective statistics – specifically mean – and compared against that of the Set A Models. We then compute the corresponding p-value based on the number of times in the re-sampled data, the mean statistic fall to the the corresponding mean of the group of Set A model/realization. These are detailed in the methodology section and correspondingly cited in the method section. Moreover, we have also re-polished this section in the revised manuscript (Line 503–511).

*Fig.S13 e,f: Try not to saturate the colorbar, it is quite difficult to read and compare.
)*

Response #10: Thank you for this suggestion. We have now try to not saturate the colorbar – and thereby reducing the sharpness of the figure as well.

References

- [1] Liu, Z., Gao, T., Zhang, W. & Luo, M. Implications of the Pacific meridional mode for summer precipitation extremes over China. *Weather Clim. Extremes* **33**, 100359 (2021).
- [2] Liu, Z., Gao, T., Zhang, W. & Luo, M. Implications of the pacific meridional mode for summer precipitation extremes over china. *Weather and Climate Extremes* **33**, 100359 (2021).